

# Seasonal carbon fluxes from vegetation and soil in a Mediterranean non-tidal salt marsh

Lorena Carrasco-Barea[1], Dolors Verdaguer[1], Maria Gispert[2], Xavier D. Quintana[3], Hélène Bourhis[1], Laura Llorens[1]

[1]Group of Soils and Vegetation in the Mediterranean (SOLIPLANT), Plant Physiology Unit, Department of Environmental Sciences, Faculty of Sciences, University of Girona, Campus Montilivi, 17003 Girona, Spain
[2]Group of Soils and Vegetation in the Mediterranean (SOLIPLANT), Soil Science Unit, Department of Chemical and Agricultural Engineering and Agrifood Technology, University of Girona, Campus Montilivi, 17003 Girona, Spain
[3]GRECO, Institute of Aquatic Ecology, University of Girona, Campus Montilivi, 17003 Girona, Spain

*Correspondence to*: Lorena Carrasco-Barea (lorena.carrasco.barea@gmail.com)

**Abstract.** Salt marshes are important ecosystems for carbon sequestration. However, while studies of atmospheric carbon exchange fluxes have been broadly performed in tidal salt marshes, they are scarce in non-tidal salt marshes. In this study we measured, throughout one year, instantaneous net $CO_2$ exchange rates from four halophytes (*Sarcocornia fruticosa, Halimione portulacoides, Elytrigia atherica* and *Salicornia patula*), which are dominant species of their corresponding habitat (an halophilous scrub, a salt meadow and a glasswort sward) of a Mediterranean non-tidal salt marsh. Soil $CO_2$ and $CH_4$ fluxes from these habitats were also measured. *E. atherica*, a perennial herbaceous species, showed the highest photosynthetic rates during the entire year, but *S. patula*, an annual succulent herb, had also remarkable photosynthetic rates in summer. Interestingly, the woody fraction of the two perennial shrubs, *S. fruticosa* and *H. portulacoides*, showed $CO_2$ uptake during most of the daily measurements. Regarding the studied habitats, the halophilous scrub and the salt meadow showed higher soil $CO_2$ emissions than the glasswort sward, being these values, in general, higher than those reported for tidal salt marshes. Both soil absorption and emission of $CH_4$ were detected. In particular, $CH_4$ emissions were remarkably high, similar to those found in low-salinity marshes, and, in general, higher than those reported for salt marshes with a high water table salinity. Soil mineralization quotients of the halophilous scrub and the salt meadow were lower than those measured at the glasswort sward, suggesting a higher soil carbon sequestration potential of the first two habitats.

## 1 Introduction

In the last decades there has been a continuous increase of atmospheric carbon dioxide ($CO_2$) and methane ($CH_4$) concentrations, two significantly active greenhouse gases, which has contributed to global climate change. In 2022, the global averages of atmospheric $CO_2$ and $CH_4$ concentrations were $417.9 \pm 0.2$ ppm and $1913.9 \pm 0.4$ ppb, respectively (Lan et al., 2023b, 2023a), which represents an increase of 150% and 264%, in relation to the atmospheric concentration of these gases in 1750 (Blunden and Arndt, 2019). In this context of continuous global warming, ecosystems play an important role



in global climate regulation, being thus essential to determine net emissions of greenhouse gases of ecosystems to estimate their effects on global warming (Ciais et al., 2013).

In particular, salt marshes play an important role in global climate regulation, since they are considered one of the most powerful carbon sinks on Earth (Laffoley & Grimsditch, 2009) due to their high net primary productivity and low rates of organic matter decomposition (McLeod et al., 2011; Mitsch and Gosselink, 2015). Regarding net primary productivity, previous studies on the photosynthetic capacity of salt marsh halophytic species have mainly focused on the effect of salinity on photosynthetic rates, being these studies mostly performed under controlled conditions (Davy et al., 2006; Duarte et al., 2014; Kuramoto and Brest, 1979; Nieva et al., 1999; Pearcy and Ustin, 1984; Redondo-Gómez et al., 2007) and less frequently under field conditions (Drake, 1989; Maricle and Maricle, 2018; Warren and Brockelman, 1989). Among the latter, the few studies that have characterized temporal patterns of $CO_2$ exchange from salt marsh plant species have been carried out in tidal salt marshes (Antlfinger and Dunn, 1979; Das Neves et al., 2008; Nieva et al., 2003).

In salt marshes, species with different photosynthetic carbon metabolism (such as $C_3$ and $C_4$ species) and/or belonging to different plant classes (such as monocotyledonous and dicotyledonous species) share the same habitats contributing differently to the photosynthetic removal of atmospheric $CO_2$ (Nieva et al., 1999; Pearcy and Ustin, 1984). Photosynthetic rates also depend on abiotic factors, such as light, temperature, flooding regime, salinity or nutrient availability (Drake, 1989; Huckle et al., 2000), being in general assumed that the highest plant photosynthetic activity occurs during the hours of the day with the highest solar radiation (midday) (Antlfinger and Dunn, 1979; Nieva et al., 2003). However, in salt marshes subjected to Mediterranean climate, high temperature and low soil moisture can become noticeable limiting factors for photosynthesis at midday, especially during summer (Das Neves et al., 2008). Other factors, such as tidal regime or soil salinity, can also reduce $CO_2$ uptake and influence the seasonal photosynthetic patterns of salt marsh species (Das Neves et al., 2008; Nieva et al., 2003).

In addition to vegetation, soil carbon fluxes also influence the role that salt marshes play in climate regulation (Bridgham et al., 2006; Chmura et al., 2003). Soil carbon ($CO_2$ and $CH_4$) emissions are related with the organic matter decomposition performed by microorganisms (Chmura, 2011), being usually positively correlated with soil organic carbon content (Li et al., 2019; Wang et al., 2016). In salt marshes, flooding also has a major effect on $CO_2$ and $CH_4$ emissions, since determines which process, aerobic respiration or anaerobic metabolism, prevails. Under aerobic conditions, organic matter can be oxidized completely to $CO_2$, whereas during flooding periods, when soil reach anoxic conditions, aerobic respiration is replaced by fermentation and methanogenesis (Mitsch and Gosselink, 2015). In this sense, ecosystems that usually or periodically have submerged soils, such as salt marshes, are among the major sources of $CH_4$ (Dalal and Allen, 2008). Nevertheless, in general, soil $CH_4$ emissions are negatively affected by salinity (Bartlett and Harriss, 1993; Livesley and Andrusiak, 2012; Poffenbarger et al., 2011), since in saline environments sulphate-reducing bacteria use to compete with methanogens for energy sources, and consequently disfavor and even inhibit methane production (Bartlett et al., 1987; DeLaune et al., 1983). Temperature also affects soil carbon emissions in salt marshes, with the highest $CO_2$ and $CH_4$ emissions being mostly recorded in the warmest season, since high temperatures enhance metabolic activity of soil




microorganisms (Chen et al., 2018; Hu et al., 2017; Wang et al., 2016). However, despite the importance that soil carbon fluxes can potentially have in climate regulation, few studies have characterized these fluxes in Mediterranean salt marshes (Wang, 2018), and, to our knowledge, no one has been performed in non-tidal salt marshes (tides range from 0.1 to 1 m, in contrast to 1-10 m of tidal salt marshes) of the Mediterranean Basin. Hence, considering the extensive coverage of non-tidal salt marshes in the Mediterranean Basin, and acknowledging the potential variations in carbon flux patterns due to distinct

flooding regimes, temperature fluctuations, and annual rainfall distribution (Ibañez et al., 2000), it is essential to study $CO_2$ and $CH_4$ fluxes in these salt marsh ecosystems.

The aim of this study was to assess the $CO_2$ fluxes from vegetation, as well as $CO_2$ and $CH_4$ fluxes from the soil, in the three main habitats of a Mediterranean non-tidal salt marsh. To achieve this objective, we measured seasonally, throughout one year, instantaneous net $CO_2$ exchange rates from the dominant plant species of these three habitats at four

daily periods (after sunrise, at midday, before sunset and at night) in green and woody (if present) tissues. Additionally, daily $CO_2$ and $CH_4$ fluxes from soils (or water, in the case of flooded soils) were also recorded.

## 2. Materials and Methods

### 2.1 Study zone

The study was performed at La Pletera, a coastal Mediterranean non-tidal salt marsh located in the north of the river Ter

mouth in the municipality of Torroella de Montgrí (Girona, NE of the Iberian Peninsula, 42°1'51"N 3°11'33"E). The largest area of this salt marsh is dominated by three Habitats of Community Interest (HCI), which, in accordance with the European Union's Habitats Directive (Council Directive 92/43/EEC, see Annex 1 of the Directive, European Commision, 1997), are habitats with a high ecological value that are at risk of disappearance. These habitats are the Mediterranean halophilous scrub (HCI code 1420), dominated by *Sarcocornia fruticosa* (syn. *Arthrocnemum fruticosum*), the Mediterranean salt

meadow (HCI code 1410), dominated by *Elytrigia atherica* (syn. *Elymus pycnanthus*) and *Halimione portulacoides* (syn. *Atriplex portulacoides*) and the glasswort sward (HCI code 1310), dominated by *Salicornia patula*, being all these species $C_3$. *S. fruticosa* and *H. portulacoides* are perennial halophytic species from the Chenopodiaceae family. *S. fruticosa* is a shrub with highly reduced leaves and succulent photosynthetic green and woody stems, while *H. portulacoides* is a creeping subshrub with slightly fleshy leaves and woody stems. *S. patula* belongs also to the Chenopodiaceae family, being an annual

succulent herb with highly reduced leaves and succulent articulated green stems. Finally, *E. atherica* is a perennial, herbaceous, monocotyledonous species belonging to the Poaceae family.

La Pletera salt marsh has a coastal Mediterranean climate with the lowest temperatures occurring in winter and the highest in summer, and with autumn and spring being the seasons with the highest monthly accumulated rainfall (Pascual, 2022). Astronomical tides are almost imperceptible (0.2-0.3 m). Meteorological events, like strong easterly winds and

rainfall, are the main cause of flooding, and mostly occur in autumn (Pascual and Martinoy, 2017).



## 2.2. Seasonal CO₂ fluxes from vegetation

Throughout one year (2017), instantaneous net $CO_2$ exchange rates (NER) were seasonally monitored for *S. fruticosa*, *H. portulacoides*, *E. atherica* and *S. patula*. Measurements were performed in green and woody plant tissues (except for *E. atherica* and *S. patula*, which had only green tissues) using a PLC3 conifer leaf cuvette (80 x 40 mm) connected to an infrared gas analyser (IRGA; CIRAS-II, PPsystems USA). Only woody stems of maximum 3 mm of diameter were used, which represented 35% and 100% of the total woody live biomass of *S. fruticosa* and *H. portulacoides*, respectively (Carrasco-Barea et al., unpublished data), since thicker stems did not allow closing the PLC3 leaf cuvette. Daylight measurements were always carried out on sunny days in sun-exposed vegetal tissues after sunrise, at midday, and before sunset. They were performed every month and a half for green tissues and every three months for woody tissues. Night measurements were taken one hour after the complete absence of light once per season (Table S1).

Plant fractions used to measure $CO_2$ fluxes were collected and stored in a fridge until sampled area was determined in the laboratory. To quantify photosynthetic area, it was considered that only half of the stem and one side of leaves received direct sunlight inside the leaf cuvette during daytime measurements, accordingly with Redondo-Gómez et al. (2007, 2010). However, for nighttime measurements, the entire stem and both sides of leaves were taken into consideration. Instantaneous NER was expressed as µmol $CO_2$ m$^{-2}$ s$^{-1}$, where m$^{-2}$ refers to tissue area. Stomatal conductance values ($g_s$) were also obtained when NER were measured, and intrinsic water-use efficiency (iWUE) of green tissues was calculated for midday measurements as the ratio between photosynthetic rates and $g_s$.

## 2.3. Seasonal carbon (CO₂ and CH₄) fluxes from soil

### 2.3.1. Soil CO₂ measurements

Measurements of soil $CO_2$ fluxes were performed during 2017, every month and a half, using the soda lime method (Edwards, 1982), which is based on the capacity of the soda lime to absorb $CO_2$. In the field, five static opaque chambers per habitat (PVC cylinders of 11 cm of diameter and 13 cm of height) were inserted 5 cm into the soil (Fig. S1). At midday of each sampling day, an open glass vessel containing soda lime, previously oven dried at 105 °C and weighed, was placed inside each chamber, and then chambers were immediately closed. After approximately 24 h, the glass vessel was hermetically sealed, collected, oven dried at 105 °C and weighted. The $CO_2$ absorbed by the soda lime was calculated by multiplying the weight gain by 1.69 as a water correction factor (Emran et al., 2012; Grogan, 1998). Daily soil respiration rates (SR; g $CO_2$ m$^{-2}$ d$^{-1}$) were calculated as follows:

$$SR = \frac{(SL_f - SL_i) * 1.69}{A * t}$$

where $SL_f$ is the soda lime dry weight (in g) after 24h in the field; $SL_i$ is the initial soda lime dry weight (in g) before being placed in the field; $A$ is the soil surface area within the chamber (m²), and $t$ is the time (in days) that soda lime remained in the field.



When the soil was flooded, taller opaque chambers were used (Fig. S1), and soil $CO_2$ fluxes were measured by collecting air samples from inside the chamber after 24h of being hermetically closed (Table S2). The $CO_2$ concentration of these samples was analyzed by gas chromatography at the Laboratory of Chemical and Environmental Engineering of the

Research Technical Services of the University of Girona (UdG). Control air samples were also taken just before closing the chambers. Daily soil respiration rates measured by gas chromatography (SR; g $CO_2$ m$^{-2}$ d$^{-1}$) were calculated as follows:

$$SR = \frac{(W_f - W_i)}{A * t}$$

where $W_f$ is the amount of $CO_2$ (grams) in the air inside the chamber after 24h of being closed, $W_i$ is the initial amount of $CO_2$ (grams) in the air inside the chamber before being closed, $A$ is the soil surface area within the chamber (m$^2$), and $t$ is the

time (in days) that the chamber remained closed. $W_f$ and $W_i$ were estimated from volumetric concentration (%) considering the air volume inside the chamber in each sampling date.

Gas chromatography analyses were not used to estimate soil respiration when the soil was not flooded because the soda-lime method is considered more reliable, since it has been observed that gas chromatography can underestimate $CO_2$ emission rates by up to 45% in comparison to other methods, such as the soda-lime (Lou and Zhou, 2006).

**2.3.2. Soil CH$_4$ measurements**

Methane fluxes between the atmosphere and the soil surface (or the water surface when the soil was flooded) were estimated using the same opaque chambers used to measure soil $CO_2$ fluxes. After 24h of being hermetically closed, air samples from inside the chambers were collected and CH$_4$ concentration was analyzed by gas chromatography at the Laboratory of Chemical and Environmental Engineering of the UdG Research Technical Services. Control air samples were also taken just

before closing the chambers.

Daily soil methane fluxes (SMF; g CH$_4$ m$^{-2}$ d$^{-1}$) were calculated as follows:

$$SMF = \frac{(Wm_f - Wm_i)}{A * t}$$

where $Wm_f$ were the grams of CH$_4$ in the air inside the chamber after 24h of being closed; $Wm_i$ were the initial grams of CH$_4$ in the air inside the chamber before being closed; $A$ was the soil surface area within the chamber (m$^2$); and $t$ was the time (in

days) that the chamber remained closed. $Wm_f$ and $Wm_i$ were estimated from volumetric concentration (%), considering the air volume inside the chamber in each sampling date.

Soil CH$_4$ and $CO_2$ fluxes (with soda lime or gas chromatography) were measured on the same days. When soil was flooded, the same air samples were used to estimate both, $CO_2$ and CH$_4$, fluxes.

**2.3.3. Carbon mineralization quotient**

The carbon mineralization quotient (Q$_{min}$) represents the fraction of soil organic carbon (SOC) mineralized in a given period of time (Pinzari et al., 1999) being thus considered a carbon sequestration index. Specifically, it represents the carbon that is



emitted to the atmosphere under inorganic forms ($CO_2$ and $CH_4$) in relation to the carbon that is stored in the soil (SOC) for a certain depth. The soil depth considered was the first 20 cm, since it was previously determined that most SOC was stored there (Carrasco-Barea et al., 2023).

Daily carbon mineralization quotients ($Q_{min}$) were calculated for each sampling date following this equation:

$$Q_{min} = \frac{C\_CO_2 + C\_CH_4}{SOC}$$

were $C\_CO_2$ represents the carbon emitted as $CO_2$ per gram of soil (considering the soil's bulk density) (mg C g soil$^{-1}$ d$^{-1}$); $C\_CH_4$ represents the carbon emitted as $CH_4$ (mg C g soil$^{-1}$ d$^{-1}$) and $SOC$ was the soil organic carbon of the first 20 cm of depth (mg SOC g soil$^{-1}$). SOC values were taken from previous measurements performed in July 2015 and 2016 in the same
experiment, after observing that these values exhibited stability and remained constant over the studied years (Carrasco-Barea et al., 2023).

### 2.4. Environmental measurements

For each sampling date and measurement location, the following soil environmental parameters were monitored: soil temperature (Ts) for the first 12 cm of depth (Digital Portable Thermometer AI 368, Acez; Singapore), soil volumetric water
content (VWC) by means of a 20 cm rod (FieldScout TDR 300 soil moisture meter, Spectrum technologies Inc; USA), and soil electrical conductivity (EC) (conductivity meter 254, CRISON instruments; Spain). Since the high EC values of these soils could affect VWC measurements, VWC values were corrected by obtaining calibration curves between TDR readings and the soil water content measured directly by determining soil weight loss over time (and considering the soil bulk density to convert data into volumetric content) in undisturbed soil samples.
Climatic data (maximum and minimum air temperature, air relative humidity and vapor pressure deficit) were obtained from l'Estartit meteorological station (Pascual, 2022), located at 2.5 km from La Pletera salt marsh.

### 2.5. Data analyses and statistics

To evaluate whether the studied plant species differed in their instantaneous net $CO_2$ exchange rates (NER), stomatal conductance and intrinsic water-use efficiency, two-way ANOVAs were performed for each sampled time of the day, using
species and sampling day as fixed factors. When the interaction between factors was significant, differences among species (for each sampling day) and sampling days (within each species) were evaluated by means of one-way ANOVAs.

To evaluate differences among habitats in soil respiration (SR), methane fluxes (SMF), carbon mineralization quotients ($Q_{min}$), soil temperature (Ts), electrical conductivity (EC) and volumetric water content (VWC), mixed models were performed using habitat as fixed factor, sampling day as repeated factor (repeated measures) and plots as random
factor. The interaction between habitat and sampling date was also included. When this interaction was significant, differences among habitats were tested for each sampling day by means of one-way-ANOVAs, except for March and




December, since, in these months, the glasswort sward had only two non-flooded plots, and, therefore, non-parametric Kruskall-Wallis tests were used.

Correlations were performed between: a) midday NER of the perennial species (*S. fruticosa*, *H. portulacoides* and *E. atherica*) and edaphic (VWC and EC) and climatic (maximum air temperature, air relative humidity and vapor pressure deficit) parameters, b) night NER and minimum air temperature, c) soil carbon fluxes (SR, SMF) and edaphic parameters (VWC, EC and Ts) measured during the entire year, and d) soil carbon fluxes of July (SR, SMF) and soil organic carbon (SOC) and total nitrogen content (TN) measured previously in July 2015 and 2016 in the same experiment (Carrasco-Barea et al., 2023). Pearson's correlation tests were usually applied, although the Spearman rank correlation coefficient was used when data did not follow a normal distribution.

The Shapiro-Wilk test was used to check the normality of data, while the Levene's test was applied to evaluate the homogeneity of variances. When factors (species, sampling day and/or habitat) were significant, Tukey's HDS *post-hoc* tests or pairwise comparisons (when data did not accomplish the assumptions of normality and/or homoscedasticity) were performed. For all the statistical tests, the significance level considered was $p < 0.05$. Statistical analyses were performed using SPSS software (IBM SPSS statistics, Corporation, Chicago, USA).

## 3. Results

### 3.1. Seasonal $CO_2$ fluxes from vegetation

Differences among species in instantaneous net $CO_2$ exchange rates (NER) from green tissues depended on the time of the day and the sampling day. Specifically, *E. atherica* had the highest photosynthetic rates (negative NER) in March and April after sunrise (with no significant differences with *S. fruticosa* in April; Fig. 1a), and throughout all the year at midday, except in June and July, when *S. patula* showed the highest values (Fig. 1c), corresponding to its growth period. Before sunset, the highest photosynthetic rates were recorded for *E. atherica* and *S. fruticosa* in February and March, for *S. patula* in June, and again for *E. atherica* during the rest of the year (Fig. 1e). At night, the highest respiration values (positive NER) were found in August and November for *E. atherica* and *S. fruticosa* (Fig. 1g).

As expected, the maximum photosynthetic activities of the green tissues of the three perennial species (*S. fruticosa*, *E. atherica* and *H. portulacoides*) were recorded in spring (March or April; Fig. S2). Moreover, the four species studied showed, in general, the highest photosynthetic rates at midday, although differences with the other times of the day were not always significant (Fig. 1 and S2). In fact, seasonal patterns of photosynthetic activity were similar at the three times of day sampled for *S. fruticosa* and *S. patula*, while, in the case of *E. atherica* and *H. portulacoides*, midday patterns differed from those measured after sunrise and before sunset (Fig. S2).

Interestingly, thin woody stems of *S. fruticosa* and *H. portulacoides* had net $CO_2$ uptake during all the light sampling times and days measured except in November, with both species showing similar values throughout the year after





sunrise and at midday (Fig. 1b, d). Before sunset, *H. portulacoides* generally had lower values than *S. fruticosa* (Fig. 1f), while, at night, *H. portulacoides* showed lower respiration values than *S. fruticosa* in November, but higher in June (Fig. 1h).

No significant correlations were found between midday NER and soil VWC, soil EC, maximum air temperature, air relative humidity or air vapor pressure deficit as well as between night NER and minimum air temperature for any of the three perennial species studied (results not shown).







**Figure 1. Instantaneous net CO$_2$ exchange rates (NER) of the four studied species after sunrise (a, b), at midday (c, d), before sunset (e, f) and at night (g, h), for green and woody tissues. Negative values indicate net photosynthetic activity, while positive values indicate net respiration. Bars represent standard errors (n = 4 after sunrise, before sunset and at night, and n = 6 at midday). Significant $p$-values for the species and sampling date factors and their interactions (according to the two-way-ANOVA results) are also shown. Asterisks indicate significant differences among species in each sampling date ($p < 0.05$), and they are depicted only when the interaction between species and sampling date was significant. NS: Not significant.**

### 3.1.1. Seasonal stomatal conductance and intrinsic water-use efficiency of green tissues at midday

During most of the year, *E. atherica* showed the highest values of stomatal conductance (g$_s$) at midday, while *H. portulacoides* had the lowest (Fig. 2a). In July and October, no significant differences in g$_s$ were found among the four species studied. In regard to intrinsic water-use efficiency, differences among species were significant only in September, when *S. fruticosa* showed lower values than the other three species (Fig. 2b).

      Concerning seasonal patterns, the three perennial species presented the highest g$_s$ values in spring (March or April), while no significant differences were observed among sampling dates for *S. patula*. iWUE values were generally higher in June and October for the three perennial species, although, due to the high variability found, differences with most of the other dates were not significant. In the case of *S. patula*, the highest iWUE values were observed in June and July.

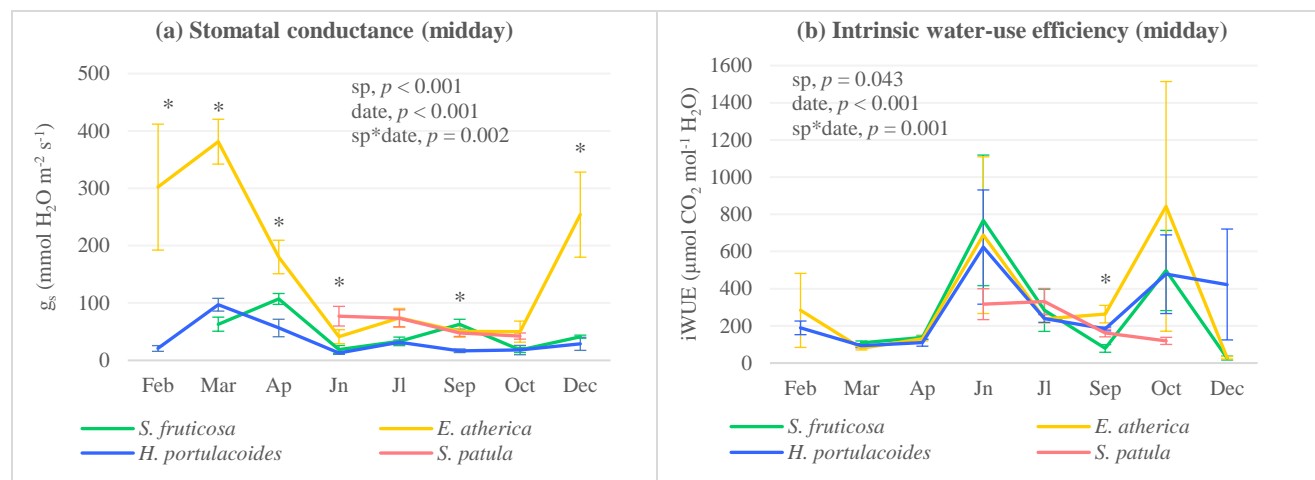

**Figure 2. Stomatal conductance (a) and intrinsic water-use efficiency (photosynthesis/stomatal conductance) (b) of green tissues of *S. fruticosa*, *E. atherica*, *H. portulacoides* and *S. patula* at midday (mean ± SE, n = 6). February values of *S. fruticosa* are missing because g$_s$ values were too low to be properly determined. Significant $p$-values for the species and sampling date factors and their interactions (according to the two-way-ANOVA results) are also shown. Asterisks indicate significant differences among species within each sampling day ($p < 0.05$).**





### 3.2. Seasonal carbon fluxes from soil

### 3.2.1. Soil environmental parameters

The highest soil temperatures (Ts) were registered during late spring and summer (Fig. 3a), in agreement with the highest air temperatures recorded during these months (Pascual, 2022). From July to December, soil temperatures were higher in the glasswort sward than in the two other habitats. Significant differences in the seasonal volumetric water content (VWC) of the

soil were only detected for the halophilous scrub, with the highest values being found in October and the lowest in March and June (Fig. 3b). No significant differences in VWC were found among habitats. Nevertheless, soil electrical conductivity (EC) was significantly higher in the glasswort sward, followed by the halophilous scrub and the salt meadow, which was the least saline (Fig. 3c). Overall, the highest values of EC were recorded in June, although these values did not differ significantly from those obtained in July and September.


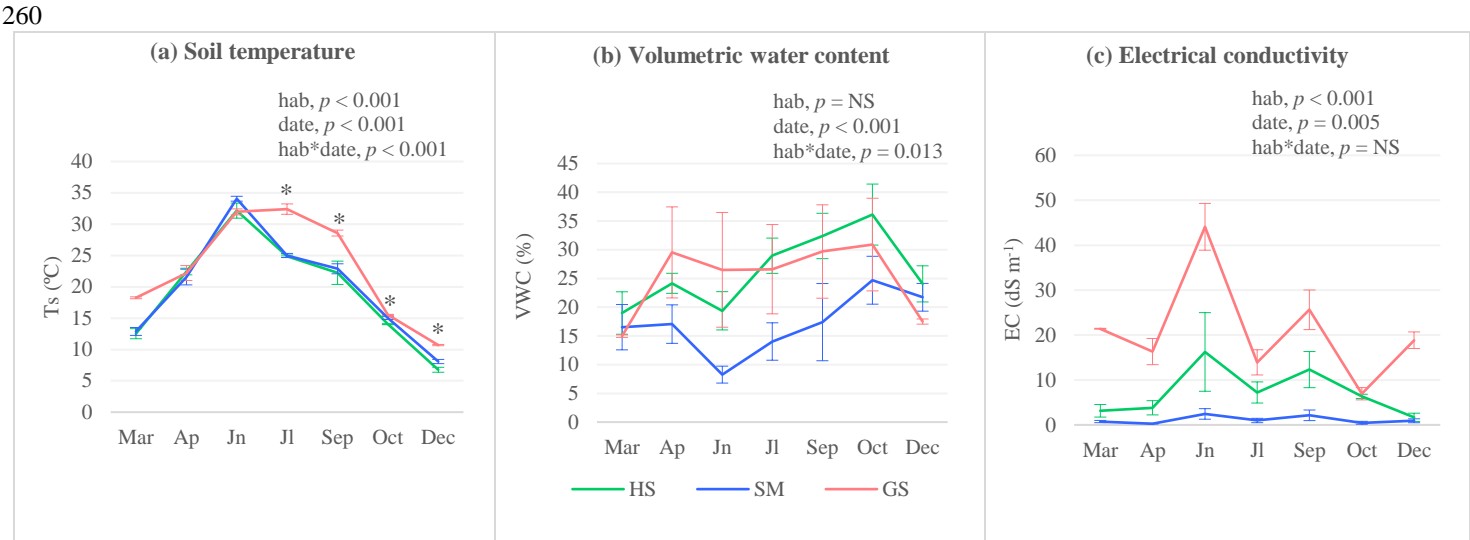

**Figure 3. Soil temperature (a), volumetric water content (b) and electrical conductivity (c) of the plots where soil carbon fluxes were measured for each sampling date and habitat. Only non-flooded plots are considered. HS: Halophilous scrub; SM: Salt meadow; GS: Glasswort sward. Bars represent ± standard errors (n = 5, except in the glasswort sward in March and December in**
**which n = 2). Significant *p*-values for the habitat and date factors and their interactions (according to the two-way-ANOVAs results) are also shown. Asterisks indicate significant differences among habitats within a sampling date (*p* < 0.05), being depicted only when the interaction between habitat and sampling date was significant. Despite the interaction between habitat and sampling date was significant for VWC, the analyses of the differences among habitats within each sampling date did not report significant results. NS: Not significant.**


### 3.2.2. Soil CO₂ and CH₄ fluxes and soil carbon mineralization quotient

Daily soil respiration (SR) for non-flooded soils of the three salt marsh habitats ranged from $4.0 \pm 0.03$ to $19.4 \pm 0.9$ g CO₂ m⁻² d⁻¹, with the highest values being recorded in July and the lowest in October and December (Fig. 4a). On the contrary,





$CO_2$ emissions were remarkably lower when soils were flooded. Regarding the differences among habitats, the halophilous

scrub and the salt meadow showed significantly higher SR values than the glasswort sward.

Remarkably high peaks of soil $CH_4$ emissions were recorded in the three habitats, despite negative values (indicating net $CH_4$ consumption) were also observed (Fig. 4b). In the halophilous scrub, soil $CH_4$ emissions were detected in April, June (with high values) and September, with the highest $CH_4$ absorption being observed in February, when the soil was flooded. In the salt meadow and the glasswort sward, soil $CH_4$ absorption was only detected in October. Maximum soil

$CH_4$ emissions were recorded in July for the salt meadow, and in March or June in flooded and non-flooded soils, respectively, for the glasswort sward.

For the three habitats, the highest soil carbon mineralization quotient ($Q_{min}$) was found in July and the lowest in December (Fig. 4c). The glasswort sward presented the highest $Q_{min}$, while the halophilous scrub and the salt meadow showed similar values.

SR and daily soil methane fluxes (SMF) were positively correlated with soil temperature (Pearson's correlation coefficients (r) 0.568 and 0.557, respectively; $p < 0.01$), while SR of July was positively correlated with SOC (r = 0.997, p = 0.045) and TN (r = 0.999, p = 0.026).





**Figure 4. Daily soil CO₂ (a) and CH₄ (b) fluxes, and soil carbon mineralization quotient (c). HS: Halophilous scrub, SM: Salt meadow, GS: Glasswort sward. Bars represent ± standard errors (n=5). In February, plots from the three habitats were flooded.**






**In March and December, the glasswort sward had flooded (n=3) and non-flooded (n=2) plots. Significant *p*-values for the habitat and sampling date factors and their interactions (according to the two-way-ANOVA results) are also shown. Asterisks indicate**
**significant differences among habitats within a sampling date (p < 0.05). NS: Not significant.**

## 4. Discussion

### 4.1. Carbon fluxes from vegetation

*Elytrigia atherica,* an herbaceous monocotyledonous species which is the dominant species of the salt meadow (Carrasco-
Barea *et al.* 2023), presented the highest photosynthetic rates during most of the year, reaching 29 µmol $CO_2$ m$^{-2}$ s$^{-1}$ in March at midday, being only surpassed by the annual species *Salicornia patula* in June/July. *E. atherica*, as a Poaceae species, has dumbbell-shaped stomata, while the other two perennial species studied, *Sarcocornia fruticosa* and *Halimione portulacoides*, have kidney-shaped stomata. Dumbbell-shaped guard cells are surrounded by subsidiary cells that participate in the pore movements through physical interaction with guard cells, allowing a faster response and a wide pore aperture (Franks and
Farquhar, 2007; Grantz and Zeiger, 1986). These conditions usually result in higher photosynthetic rates, especially in fluctuating environments such as salt marshes, when a fast stomatal response is an advantage for the photosynthetic process (Chen et al., 2017; Franks and Farquhar, 2007). In accordance with its higher photosynthetic rates, *E. atherica* also showed greater stomatal conductance values during almost the entire study period and higher leaf carbon concentration (Carrasco-Barea et al., 2023) compared to the other two perennial species. Previous studies in salt marshes have also reported higher
net photosynthetic rates in monocotyledonous grasses compared to succulent Chenopodiaceae species (Kuramoto and Brest, 1979; Nieva et al., 1999; Pearcy and Ustin, 1984).

In the case of *S. fruticosa*, the maximum mean photosynthetic rate recorded at La Pletera salt marsh was 14.3 ± 0.8 µmol $CO_2$ m$^{-2}$ s$^{-1}$, which was higher than those obtained in other studies, where maximum values were between 3-6 µmol $CO_2$ m$^{-2}$ s$^{-1}$ (Abdulrahman and Williams, 1981; Nieva et al., 1999; Redondo-Gómez et al., 2006; Redondo-Gómez and
Mateos-Naranjo, 2010). However, these previous studies were performed with potted plants, collected from field sites and then cultivated under controlled conditions of light, temperature and soil salinity. The same occurred with *E. atherica* and *S. patula,* being their maximum photosynthetic rates at La Pletera (29.1 ± 2.4 and 20.8 ± 2.9 µmol $CO_2$ m$^{-2}$ s$^{-1}$, respectively) higher than those previously reported for *E. atherica* (18 µmol $CO_2$ m$^{-2}$ s$^{-1}$, Rozema & Diggelen 1991) and for the annual species *Salicornia ramosissima* (14 µmol $CO_2$ m$^{-2}$ s$^{-1}$, Pérez-Romero et al. 2018) grown under controlled conditions. On the
contrary, the maximum mean photosynthetic rate obtained for *H. portulacoides* at La Pletera salt marsh (9.3 ± 0.8 µmol $CO_2$ m$^{-2}$ s$^{-1}$) was slightly lower than the values previously obtained for the same species (15 to 18 µmol $CO_2$ m$^{-2}$ s$^{-1}$) growing under controlled (Redondo-Gómez et al., 2007) or field (Das Neves et al., 2008) conditions. Therefore, studies reporting photosynthetic rates of dominant salt marsh plant species under field conditions are scarce, and the values obtained often



diverge substantially from those recorded under controlled conditions. This disparity can be a serious shortcoming for the
development of accurate predictive models regarding the carbon balance of these ecosystems.

The elevated photosynthetic rates found in the most common plant species of La Pletera would explain their high
mean maximum water use efficiency values (767 ± 351, 624 ± 307, 843 ± 671 and 331 ± 70 µmol $CO_2$ mol$^{-1}$ $H_2O$ for *S.
fruticosa, H. portulacoides, E. atherica* and *S. patula,* respectively), which exceed those reported under natural conditions
for other salt marsh species, such as *Spartina densiflora* (around 100 µmol $CO_2$ mol$^{-1}$ $H_2O$; Nieva et al., 2003), *Halimione
portulacoides* and *Limoniastrum monopetalum* (around 400 µmol $CO_2$ mol$^{-1}$ $H_2O$; Das Neves et al., 2008), as well as for
*Salicornia ramosissima* under controlled conditions (around 100 µmol $CO_2$ mol$^{-1}$ $H_2O$ Pérez-Romero et al., 2018). In
general, the perennial halophytic species studied (*S. fruticosa, E. atherica* and *H. portulacoides*) presented their highest
photosynthetic activity in March and/or April, which coincides with their growing season and with favorable environmental
conditions (maximum temperatures around 18 °C; Pascual, 2022), as it was also found for a non-tidal population of *Spartina
alterniflora* located at the upper part of a tidal salt marsh in Portugal (Nieva et al. 2003). Interestingly, photosynthetic rates
of the studied species at La Pletera were much lower in autumn than in spring, despite the environmental parameters, such as
temperature and soil moisture, were also favorable to photosynthesis (especially in October, where maximum temperature
was 21°C and soil VWC was even higher than in March and April; Pascual, 2022). A possible explanation might be related
with the high accumulation of ions and soluble carbohydrates that these species would present after a salt stress period, such
as the one occurring in the Mediterranean salt marshes during summer (Gil et al., 2014, 2011; Redondo-Gómez et al., 2007).
In dicotyledonous species, high salinity conditions can induce the accumulation of Na$^+$ and Cl$^-$ in the cytoplasm (Munns,
1993) affecting photosynthesis (Almeida et al., 2017; Chaves et al., 2009). In addition, the accumulation of high intracellular
levels of soluble carbohydrates as a salt tolerant mechanism in salt marsh monocotyledonous species (Gil et al., 2013) can
promote a feedback inhibition of photosynthesis (Munns, 1993).

Thin woody tissues (stem diameter < 3 mm) of *S. fruticosa* and *H. portulacoides* also showed photosynthetic
activity, especially in March and May and before sunset, with values of photosynthesis reaching 12 µmol $CO_2$ m$^{-2}$ s$^{-1}$, in
agreement with data reported for Californian evergreen species (Saveyn et al., 2010) or for savannah shrubs and trees
(Cernusak et al., 2006; Levy and Jarvis, 1998). As occurs with the woody stems of these other species, the thin woody stems
of *S. fruticosa* and *H. portulacoides* plants growing at La Pletera salt marsh presented a layer of green cells under the
periderm (Fig. S3). This layer of cells would likely be responsible for the recorded photosynthesis by using the light that
passes through the thin bark and the $CO_2$ that penetrates through lenticels. Taking into account that woody stems are
typically overlooked in studies focused on characterizing $CO_2$ exchange in vegetation, present results highlight the
importance of measuring and incorporating the photosynthetic activity of thin stems into such investigations. This is
particularly crucial for ecosystems like the Mediterranean salt marshes, dominated by succulent Chenopodiaceae species, as
those of the genera *Sarcocornia,* in which the woody fraction represents a significant fraction of the plant aerial biomass
(Carrasco-Barea et al., 2023).





Regarding night respiration rates, the highest values for the four species were recorded in summer (August) and/or autumn (November), being especially elevated those found for the green tissues of *S. fruticosa* and *E. atherica* during these months. Night respiration of these species seems not to be directly affected by air temperature, since correlations between night NER

and minimum air temperature were not significant for any of the three perennial species. In *E. atherica* and *S. fruticosa,* the high values of night respiration registered in August coincide with their flowering period and, thus, with a high energetic demand (Bustan and Goldschmidt, 1998; Lambers et al., 2008). In the case of *S. fruticosa,* flowers, which are very abundant and small, remained inside the chamber during the measurements, and this could have contributed to increase respiration rates in August (Bustan and Goldschmidt, 1998). In November, respiration rates were also very high despite the

minimum temperature was much colder (4.6 °C) than in August (22.2 °C) and similar to February (5.9°C) (Pascual, 2022). High night respiration values in autumn might be explained, at least partially, by the accumulation of soluble carbohydrates and/or by the increase in the chloroplast redox status (Atkin et al., 2005; Koch, 1996) due to low temperatures, although more research is needed to clarify this.

**5.2.2. Carbon fluxes from soil**

Soil respiration (SR) values recorded at the halophilous scrub and the salt meadow were higher than those found at the glasswort sward. These differences might be related to the soil C and/or N content, which can affect the microorganism-mediated soil organic matter decomposition (Gougoulias et al., 2014; Oertel et al., 2016). Positive correlations between $CO_2$ emissions and soil organic carbon (SOC) or total nitrogen (TN) have been found for coastal saline soils (Li et al., 2019;

Wang et al., 2016), as well as for soils of other terrestrial ecosystems (Merbold et al., 2011; Shi et al., 2014). Accordingly, a positive correlation was found between July SR and SOC or TN content at the halophilous scrub and the salt meadow of La Pletera salt marsh, since these two habitats had higher content of SOC and TN than the glasswort sward (Carrasco-Barea et al., 2023). Previous studies have shown that most of the $CO_2$ produced during decomposition is derived from organic material recently incorporated to the soil, with only a small fraction (approx. 10%) of soil respiration being derived from

decomposition of older, more recalcitrant, carbon compounds (Giardina et al., 2004; Trumbore, 2000). Considering this, the lower soil $CO_2$ emissions recorded in the glasswort sward would be in accordance with its more humified and stable soil organic matter (Gispert, unpublished data) and the scarce amount of litter found in this habitat (Carrasco-Barea et al., 2023). In addition, root respiration could have contributed to the high soil respiration found in the halophilous scrub and the salt meadow habitats, since *S. fruticosa*, *H. portulacoides* and *E. atherica* have superficial root systems. On the contrary, the

sparse vegetation (which is only alive during few months) of the glasswort sward and the poorly developed root system of its dominant species, *S. patula*, would make negligible the contribution of roots to soil respiration in this habitat.

Daily soil respiration presented a similar seasonal pattern in the three habitats, with the highest values being recorded in summer (July), which agrees with the positive correlation found between SR and soil temperature. These





findings are consistent with numerous previous studies in which the highest soil $CO_2$ emissions are produced in the warmest
season, since high temperatures enhance metabolic activity of soil microbes (Chen et al., 2018; Hu et al., 2017; Wang et al.,
2016). In general, our estimations of daily soil $CO_2$ emissions were in the upper part of the range previously published for
salt marshes (Table 1), except for the study of Hu et al. (2017). One relevant difference between La Pletera salt marsh and all
the salt marshes considered in Table 1, except the one studied by Hirota et al. (2007), is the daily tidal flood. In tidal salt
marshes, flooding occurs once or twice every day, while it is occasional at La Pletera. This allows an airier soil at La Pletera,
which favours microbial respiration. In fact, previous studies under field (Kathilankal et al., 2008; Moffett et al., 2010) or
laboratory (Jones et al., 2018; Wang et al., 2019) conditions support a negative effect of flooding on soil $CO_2$ emissions. At
La Pletera, the reduction of soil $CO_2$ emissions by flooding becomes evident when comparing the $CO_2$ fluxes of flooded and
non-flooded soils, although a certain underestimation of $CO_2$ emissions in flooded soils due the use of the gas
chromatography method cannot be excluded (Lou and Zhou, 2006). The study of Hirota et al. (2007), performed also in a
non-tidal salt marsh, reported daily soil $CO_2$ emissions similar to those found at La Pletera.

Methane fluxes in salt marshes are the result of $CH_4$ production, consumption and diffusion from soil anaerobic
zones to the atmosphere (Bodelier and Laanbroek, 2004; Sun et al., 2013). A strict condition for the growth of methanogen
microorganisms is the complete absence of oxygen, which is common in ecosystems where soil is periodically flooded, as
occurs in wetlands and salt marshes (Kayranli et al., 2010). Methane generated in the anoxic layers of soil diffuses to more
superficial and aerated soil layers or to the overlying water column were it can be oxidized, thereby reducing the amount of
methane that eventually reaches the atmosphere (Dean et al., 2018). At La Pletera salt marsh, net soil $CH_4$ absorption
(negative values of soil $CH_4$ flux) was found in the three habitats studied, but especially in the halophilous scrub, in
accordance with previous studies performed in other salt marshes (Bartlett and Harriss, 1993; Chen et al., 2018; Hirota et al.,
2007; Sun et al., 2013). Nevertheless, soil $CH_4$ emissions have also been recorded in all the studied habitats of La Pletera
(reaching $271 \pm 90$, $161 \pm 58$ and $110 \pm 59$ mg $CH_4$ m$^{-2}$ d$^{-1}$ in the halophilous scrub, the salt meadow and the glasswort
sward, respectively). In the glasswort sward, peaks in $CH_4$ emissions were observed both when the soil was not flooded ($110
\pm 59$ mg $CH_4$ m$^{-2}$ d$^{-1}$) and when it was flooded ($131 \pm 45$ mg $CH_4$ m$^{-2}$ d$^{-1}$), highlighting that methane oxidation in the
overlying water column would not be happening.

The highest soil methane fluxes were detected in the warmest months, being recorded in June for the halophilous
scrub and the glasswort sward, and in July for the salt meadow. The same seasonal trend has been broadly found in other salt
marsh studies, and it would be explained by the positive effect of higher temperatures on microbial activity (Bartlett et al.,
1987; Chen et al., 2018; Hu et al., 2017; Wang et al., 2016; Yuan et al., 2015). Despite no soil anaerobic conditions would be
expected during summer because of the low soil VWC at La Pletera salt marsh, anaerobic conditions could be found at the
water table level, which is around 30-50 cm depth (Menció et al., 2023). At this depth, there is still organic matter
susceptible to be decomposed (Amorós, 2018), which could promote $CH_4$ production. The $CH_4$ produced in the anaerobic
zone can easily diffuse through air-filled macropores, especially during summer when high temperatures at the soil surface
would promote soil water evaporation to the atmosphere (Denier Van Der Gon et al., 1996).




CH$_4$ emissions are, in general, negatively affected by salinity, being usually higher in freshwater wetlands than in salt marshes (Bartlett and Harriss, 1993; Hu et al., 2017; Poffenbarger et al., 2011). However, according to the Poffenbarger

et al. (2011) revision, the soil salinity threshold that would reduce CH$_4$ emissions would be around 18 ‰. Indeed, Poffenbarger et al. (2011) found significantly lower CH$_4$ emissions values ($3 \pm 5$ mg CH$_4$ m$^{-2}$ d$^{-1}$) in polyhaline salt marshes (salinity >18 ‰) compared to mesohaline (salinity 5-18 ‰), oligohaline (salinity 0.5-5 ‰) and freshwater (salinity 0-0.5 ‰) ($44 \pm 30$, $411 \pm 578$ and $115 \pm 208$ mg CH$_4$ m$^{-2}$ d$^{-1}$, respectively) marshes. In the present study, we detected relatively high soil CH$_4$ emissions, similar to those reported for freshwater marshes by Poffenbarger et al. (2011), which could be explained

by the low salinity (0.86‰) of the water table of La Pletera (Mencić et al., 2017). According with this, some previous studies in salt marshes with a low salinity water table, such as those of Hirota et al. (2007) (2‰), Bartlett et al. (1987) (0-12‰), Hu et al. (2017) (4‰) and Wang et al. (2016) (13-21‰), also found high CH$_4$ emissions (Table 1). On the other hand, as summer progresses (from July onwards), the absence of important rainfall episodes at La Pletera salt marsh, together with the sea water intrusion, moves the saltwater wedge inland, increasing groundwater salinity until levels similar to those of the

sea (approximately 32‰) (Mencić et al., 2017). Therefore, the sharp decrease in CH$_4$ emissions recorded in July in the halophilous scrub and the glasswort sward (the closest habitats to the sea) might be the consequence of more saline conditions in the water table. This would not be the case of the salt meadow, the most distant habitat from the sea, where maximum CH$_4$ emissions were recorded in July.

Overall, soils of the halophilous scrub and the salt meadow showed higher carbon losses as CO$_2$ and CH$_4$ emissions

than the glasswort sward. However, when comparing the mineralization quotients among habitats, the glasswort sward had the highest values, which would indicate that soils of the halophilous scrub and the salt meadow would have a higher carbon sequestration potential, despite their higher soil carbon emissions.






**Table 1. Published data on daily soil CO₂ and CH₄ emissions from salt marshes at different locations. Values without parentheses indicate maximum values of CO₂ and CH₄ emissions, while values in parentheses indicate annual averages. A rank of mean maximum values is given when emissions have been measured in more than one area or habitat within the same salt marsh. When C emissions were measured in zones experiencing different degrees of tidal influence at the same tidal salt marsh, the term non-tidal is employed to indicate areas where flooding occurs only at some times of the year (instead of being daily flooded by tides). ND: No data.**

| Location | Tidal regime | Sampling frecuency | Climate zone | CO₂ emissions (g CO₂ m⁻² d⁻¹) | CH₄ emissions (mg CH₄ m⁻² d⁻¹) | References |
|---|---|---|---|---|---|---|
| **La Pletera salt marsh, Spain** | **Non-tidal** | **Seasonal** | **Temperate (Mediterranean)** | **13.2–19.4 (8.4–12.3)** | **109.6–270.5 (22.0–36.5)** | **This study** |
| Carpinteria salt marsh, California (USA) | Non-tidal | Seasonal | Temperate (Mediterranean) | 5.3 (3.7) | (-0.08) | Wang, (2018) |
| Carpinteria salt marsh, California (USA) | Tidal | Seasonal | Temperate (Mediterranean) | 2.8–3.8 (2.0–2.7) | (1.9–2.2) | Wang, (2018) |
| Lake Nakaumi salt marsh, Japan | Non-tidal | August | Temperate | 17.4 | 845 | Hirota et al. (2007) |
| Gulf of St. Lawrence, New Brunswick (Canada) | Non-tidal | August | Temperate | 11.6 | 0.5 | Chmura et al., (2011) |
| Bay of Fundy, New Brunswick (Canada) | Tidal | August | Temperate | 9.5 | 0.8 | Chmura et al., (2011) |
| Bay of Fundy, New Brunswick (Canada) | Tidal | July-September | Temperate | 2.3–2.8 | 0.5–3.7 | Magenheimer et al. (1996) |
| York River delta, Virginia (USA) | Tidal | Seasonal | Temperate | ND | 46-259 (5.6-22) | Bartlett et al. (1987) |
| Soenke-Nissen-Koog, Germany | Non-tidal | Seasonal | Temperate | 4.7 (2.4) | ND | Khan (2016) |
| Soenke-Nissen-Koog, Germany | Tidal | Seasonal | Temperate | 0.2 (-0.03) | ND | Khan (2016) |
| Min River estuary, China | Tidal | Seasonal | Subtropical | 84 (34) | 382 | Hu et al. (2017) |
| Jiulong River estuary, China | Tidal | Seasonal | Subtropical | 4.3 (1.1) | 480 (153) | Wang et al. (2016) |
| Yellow River delta, China | Tidal | Seasonal | Subtropical | 0.7–0.7 (0.1-0.4) | 7.2–9.6 (1.7-1.8) | Chen et al. (2018) |
| Yellow River delta, China | Tidal | Seasonal | Subtropical | ND | -9.4–12 | Sun et al. (2013) |
| Yellow River delta, China | Tidal | Seasonal | Subtropical | (0.13–0.41) | (17–18) | Chen et al. (2013) |
| Mississippi River delta, Louisiana (USA) | Tidal | Seasonal | Subtropical | 4.4–17.6 | ND | DeLaune & Pezeshki (2003) |



## 6 Conclusions

This study emphasizes the remarkable atmospheric $CO_2$ removal capacity through photosynthesis exhibited by the four dominant non-tidal Mediterranean salt marsh species studied, and especially by *Elytrigia atherica*. The green parts of these species had net $CO_2$ uptake along the day during most of the year, with the recorded values being generally higher compared with previous data for the same or similar species, which would indicate a significant potential for carbon sequestration. Besides, the thin woody stems of *Sarcocornia fruticosa* and *Halimione portulacoides* had net $CO_2$ uptake in most measurements, highlighting the importance of considering this fraction when characterizing daily and seasonal $CO_2$ fluxes from these ecosystems.

The halophilous scrub and the salt meadow had higher soil $CO_2$ emissions than the glasswort sward, and, in general, these values were higher than those reported for temperate and subtropical tidal salt marshes. Both soil $CH_4$ absorption and emission were detected, being soil $CH_4$ emissions remarkably high. In general, $CH_4$ emissions from La Pletera soils were higher than those reported for other salt marshes with high water table salinity, but similar to those found in low salinity salt marshes. Remarkably, soils from the halophilous scrub and the salt meadow presented lower mineralization quotients than those of the glasswort sward, suggesting a higher potential for carbon sequestration.

The high variability among species and habitats observed in the present study, as well as the differences between La Pletera and other salt marshes, concerning the carbon cycle highlights the importance to increase the availability of data on carbon fluxes from salt marshes. This is essential to be able to make more accurate predictions regarding carbon emissions from these ecosystems, emphasizing the importance of further field research on this subject.

## Data availability

Data obtained for carbon fluxes can be downloaded at https://dataverse.csuc.cat/, while climatic data are available at http://www.meteolestartit.cat.

## Author contributions

Lorena Carrasco-Barea: conceptualization, data curation, methodology, formal analysis, investigation, writing– original draft, writing – review and editing. Dolors Verdaguer: conceptualization, writing – review & editing, supervision. Maria Gispert: conceptualization, writing – review & editing, supervision. Xavier D. Quintana: conceptualization, funding acquisition. Hélène Bourhis: methodology. Laura Llorens: conceptualization, writing – review & editing, supervision.



**Competing interest**

The authors declare that they have no conflict of interest.

**Acknowledgements**

This work was supported by the Life+ Program of the European Commission [Life Pletera; LIFE13NAT/ES/001001]. L.C-B. held a PhD grant [IFUdG2015] from the University of Girona. We are grateful to the Parc Natural del Montgrí, les Illes Medes i el Baix Ter for the support received to perform this study.

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
