# Peer review of "Seasonal carbon fluxes from vegetation and soil in a Mediterranean non-tidal salt marsh"

_EGUsphere, 2024_

## Referee Comment (RC1)

Review for 2024-1320

Abstract

Line 14: Clarify that H. portulacoides and E. atherica are part of the same habitat (similar to line 85)

Introduction

Consider also methylotrophic methanogens which persist in saline environments.

Line 68- How extensive are non-tidal marshes in the Mediterranean? Elsewhere?

Line 85-90: Which plant species are C3 vs C4?

Line 93: Specify the temperature and salinity ranges typical of this region. Are there any salinity differences in the marsh soils between habitats or seasons?

Section 2.2

How does severing the plant stems and leaves affect the CO2 fluxes? How long were they stored in the refrigerator?

(Were any measurements done on live, intact plants in the field?)

Line 110-111: How was stomatal conductance measured?

Line 126-130: Specify make/model of the gas chromatograph- was it using flame ionization detector?

Two methods for soil respiration rates are reported: The soda lime method was used when soils were not flooded. A gas chromatography based method was used when soils were flooded. How was "flooded" defined? How many measurements were made with each method? (This information will help readers to understand whether flooding frequent or infrequent).

I am not familiar with the soda lime method. Authors should better support their statement that gas chromatography underestimates CO2 fluxes relative to the soda lime method. With only an initial and final time point, over 24 hours, the fluxes are not very precise. They may be affected by artifacts such as accumulation of pressure or altered temperatures, both of which could influence the gas fluxes measured.

Results

Figures: Colors or symbols are needed to distinguish the species represented by each line.

Figure 4a: Since these are different methods used to measure CO2 fluxes from flooded vs unflooded soils, the study should not make claims about differences in soil respiration between flooded and exposed conditions. Likewise, authors should omit the flooded data points from Figure 4a to avoid direct comparison with the non-flooded data.

Discussion

Could the higher photosynthetic rates be related to C4 metabolism in E. atherica (in addition to structural difference in stomata?) Which species are C3 vs C4?

Line 327: Listing the species in consistent order of water use efficiency would be clearer for the reader

Line 397-399: Avoid direct comparison of flooded and unflooded $CO_2$ fluxes (as mentioned above) due to differences in methods. Authors might rather consider that flooding waters are a known physical barrier to gas exchange.

Table 1: Which methods were used in the studies on this table? Are they comparable to those in this study?

Discussion of methane lines 401-414: Most of the methane fluxes were positive, and so authors should not mislead the reader by first discussing negative fluxes (indicating consumption). Similarly, the methane emissions did not differ statistically between habitats. Discussion in this section should therefore focus on what might have been similar between habitats and/or how the general magnitude of fluxes falls within the range reported in other marshes.

Line 425-435: This paragraph about salinity relationships to methane emissions is useful for readers to place this study site and its findings in context. This information about the site salinity should be incorporated into the methods/ site description.

Conclusions: Authors should discuss the possible relationship of the high $CO_2$ uptake of woody tissues with the high carbon sequestration potential (as reflected by low mineralization quotients) for the salt meadow and halophylic scrub. Can this help to reconcile the finding of lower soil mineralization quotients despite the high respiration and methane emissions observed?

$CO_2$ emissions may be higher in this study than in other previous studies due to the long period of chamber closure (24h) and associated artifacts discussed above.

---

## Referee Comment (RC2)

General comments:

In a world dealing with climate change, there is a need to better understand all ecosystems. Studies like this one, investigating GHG exchange in understudied ecosystems like non-tidal salt marches are relevant and important. The combination of in-situ $CO_2 - CH_4$ soil fluxes with $CO_2$ vegetation fluxes in the different habitats results in interesting insights and a valuable addition to the laboratory studies with controlled conditions previously carried out. The study highlights the seasonal variability  and the differences between species well.

The Materials and Methods sections could be more detailed. Many elements are not mentioned here such as soil information, salinity, more specific climate data, the amount of data points taken and details about calculations are also left out. Possible additions and suggestions are mentioned in the attached file.

The Carbon mineralization quotient is not entirely clearly explained for me in the method section and not much explanation is given in the result and conclusion section. I think more explanation is needed around the mineralization quotient calculation and some discussion is needed around the carbon sequestration potential of the habitats as this result is interesting but not well supported.

The authors mention large discrepancies between measurement methods (GC, soda lime) and between in situ and laboratory experiments, therefore  this information should be added when comparing to literature. Especially for the soil fluxes it should be mentioned which method and closure time is used in the literature you compare to. In this study, the closure time of the chamber for the GC method is very long, with only 2 data points (before and at the end of the closure time), so this will seriously influence the fluxes.

Data presentation can be improved by the inclusion of a table with soil parameters, a table with the main results, a map of the study region and graph of climate data (either in supplementary or in the main text).

More comments and also some technical corrections and suggestions for readability are included in the attached document.

**Specific comments:**

*Introduction*

Line 68: How extensive are the salt marshes worldwide/in this region and what is the proportion of tital vs non-tidal in this region?

*Materials and Methods*

Line 79: Authors could add a map of the region

Line 92: Authors could add some climate data of the region both in numbers and a graph. Annually average rainfall, mean temperatures, … also add some soil data if this is available. How many months is the soil flooded on average? Is this different between the different habitats? Be sure to add the bulk density, SOC, C and N from the three habitats as these are important in the discussion

Section 2.2: Is the NER measured in situ with the leaves and tissues attached to the plants? How many leaves and tissues were measured per plant and how many plants were measured per species per session and in total?

Section 2.3: I'm don't know the soda lime method very well, but was the amount of soda lime needed for the chambers tested to be sure of complete absorption of the efflux or was this based this on previous literature?

I also think the statement starting in line 137 should be more nuanced or better supported, as in the same reference (Lou and Zhou, 2006), there is also indicated "The method (soda lime) tends to overestimate soil $CO_2$ efflux in its low range and underestimate it in its high range compared-with dynamic methods (Yim *et al.* 2002). The technique can potentially underestimate soil surface $CO_2$ effluxes by 10 to 100% (Norman *et al.* 1992, rochette *et al.* 1992, Haynes and Gower 1995, Nay *et al.* 1994)."

The sentence used in this article is based on the sentence from (Lou and Zhou, 2006) "The GC method can potentially underestimate the rate of soil $CO_2$ fluxes in comparison with other methods by up to 45% (Knoepp and Vose 2002)." But if you continue to Knoepp and Vose 2002, it seems to me that the SODA method used in this study is not that good either compared to the other methods and underestimates the $CO_2$ emissions even more than the GC method.

In this study serious underestimation is however a possible issue as the closure time is 24h. This is a long time in which saturation in the headspace can occur. The accumulation of the gas inside the chamber can limit the further emission. However this underestimation is not linked to the GC method but rather to the closure time of the chamber.

Mention how many measurements were taken in flooded state and how many in non-flooded state and how this is different for different habitats.

*Wf* and *Wi* were estimated from volumetric concentration (%) considering the air volume inside the chamber in each sampling date.

> ➔ Is it meant here inclusion of temperature and pressure measurement from the chamber on the sampling date to transform the ppm/ppb measurements from the GC to g $CO_2$/$CH_4$?
> ➔ If yes, which temperature and pressure is used?

Line146: very small comment but the unit of SMF is g $CH_4$ m-2 d-1 here but later on the unit mg $CH_4$ m-2 d-1 is consistently used.

Line 164: The bulk density of soil is never mentioned before, so authors could add the values. Also mention the SOC values used (see also previous comment on line 92).

I'm not familiar with the carbon mineralization quotient. How were the $C\_CO_2$ and $C\_CH_4$ values calculated? How was the transformation from the unit of "g $CH_4$ m-2 d-1" and "g $CO_2$ m-2 d-1" to "mg C g soil-1 d-1"?

*Results*

Concerning the correlation between SR and SOC and TN found in July: there is one value of SOC and TN for each habitat, so three in total? Or is there a value for each plot?

*Discussion*

The authors could add a table (maybe in supplementary) with mean/min/max values of water use efficiency, photosynthetic rates, …

*Section 5.2.2.*

The soil C and N content is put forward as possible explanation in line 372 and 377 for the higher SR in HS and SM than in GS, so it think the values of SOC and TN should be added in the paper (maybe in supplementary).

Line 375-378: Accordingly, a positive correlation was found between July SR and SOC or TN content at the halophilous scrub and the salt meadow of La Pletera salt marsh, since these two habitats had higher content of SOC and TN than the glasswort sward (Carrasco-Barea et al., 2023).

> ➔ Was there only a positive correlation found between SR and SOC and TN for HS and SW because in the results this is not specified (see my one remark in the section Results above). Also if you have one value of SOC or TN per habitat, then how is a correlation per habitat found, which brings me back to my previous question in Result section?

> Maybe this needs to be rephrased, framing that the positive correlation found between July SR and SOC or TN content across the habitats together with the fact that HS and SM have higher SOC an TN underpin the statement that the differences in SR might be related to the soil C and/or N content.

Line 394: can the "occasional tide" be more specific. Are we talking about flooding during several weeks in specific months or also small occasional floods once every week?

Line 406-413: The way this section is written makes it seem like there is mainly absorption of methane and then some sudden high emission peaks, while it is actually mainly emissions that are measured. Maybe a percentage of negative fluxes to total fluxes can be given.

Line 411-413: "In the glasswort sward, peaks in CH4 emissions were observed both when the soil was not flooded (110± 59 mg CH4 m-2 d-1) and when it was flooded (131 ± 45 mg CH4 m-2 d-1), highlighting that methane oxidation in the overlying water column would not be happening."

I'm not sure that emissions during both non flooded and flooded states prove that there is no oxidation in the overlying water column. It states that the methane oxidation is rather limited or actually that the methane oxidation is not a big factor as the net emission is still large.

Line 439-442: Not much is said about this mineralization quotient. Is there an explanation for the higher sequestration potential of the HS and SM or the lower sequestration potential of GS? I assume that the SOC amount of GS is very low compared to the SOC amount of HS and SM (be sure to put these values in).

Table1. The authors mention large discrepancies between measurement methods and between in situ or laboratory experiments, therefore this information can maybe be added in the table. I would

suggest to move this table to supplementary material and instead incorporate some tables with the data gathered from this study. For readers who want to quickly scan the paper, a table with the mean results of the study would be handy.

**Technical corrections:**

*Abstract*

Line 19-21: Regarding the studied habitats, the halophilous scrub and the salt meadow showed higher soil CO2 emissions than the glasswort sward, , and the overall emissions were higher than those previously reported for tidal salt marshes.

*Introduction*

Line 29:  compared to the atmospheric concentration

Line 30-32: In this context of continuous global warming, ecosystems play an important role in global climate regulation Therefore, it is essential to determine net emissions of greenhouse gases of ecosystems to estimate their effects on global warming.

Line 34: (Laffoley & Grimsditch, 2009) is not present in the references.

Line 35-39: Previous studies on the photosynthetic capacity of salt marsh halophytic species have mainly focused on the effect of salinity on photosynthetic rates,  and most of these studies  were performed under controlled conditions (Davy et al., 2006; Duarte et al., 2014; Kuramoto and Brest, 1979; Nieva et al., 1999; Pearcy and Ustin, 1984; Redondo-Gómez et al., 2007) and less frequently under field conditions (Drake, 1989; Maricle and Maricle, 2018; Warren and Brockelman, 1989).

Line 45-47: Photosynthetic rates also depend on abiotic factors, such as light, temperature, flooding regime, salinity or nutrient availability (Drake, 1989; Huckle et al., 2000). In general it is assumed that the highest plant photosynthetic activity occurs during the hours of the day with the highest solar radiation (midday).

Line 55-56: In salt marshes, flooding also has a major effect on CO2 and CH4 emissions, since it determines which process, aerobic respiration or anaerobic metabolism, prevails.

Line 60-62: Nevertheless,  generally soil CH4 emissions are negatively affected by salinity (Bartlett and Harriss, 1993; Livesley and Andrusiak, 2012; Poffenbarger et al., 2011), since in saline environments sulphate-reducing bacteria  compete with methanogens for energy sources, and consequently disfavor and even inhibit methane production.

Line 65-68: Despite the importance  of soil carbon fluxes  in climate regulation, few studies have characterized these fluxes in Mediterranean salt marshes (Wang, 2018), and, to our knowledge, not one study has been performed in non-tidal salt marshes (tides range from 0.1 to 1 m, in contrast to 1-10 m of tidal salt marshes) of the Mediterranean Basin.

*Materials and Methods*

Line 79-80: The study was performed at La Pletera, a coastal Mediterranean non-tidal salt marsh located in the north of the river mouth of the Ter  in the municipality of Torroella de Montgrí.

Line 86-87: being all these species C3 -> , all these species being C3

Line 136: the air volume inside the chamber  on each sampling date.

Line 151:  air volume inside the chamber  on each sampling date.

Line 157 : stored in the soil (SOC)  at/above a certain depth

*Results*

Line 203: Differences among species in instantaneous  (NER) from green tissues

Line 205: after sunrise with no significant differences with *S. fruticosa* in April (Fig. 1a),

Line 232: During most of the year, *E. atherica* showed the highest values of   at midday

Line 272:  for non-flooded soils of the three

Line 273-274:  CO2 emissions were remarkably lower when soils were flooded.

Line 276-277: Remarkably high peaks of soil CH4 emissions were recorded in the three habitats,  but also negative values, indicating net CH4 consumption were  observed (Fig. 4b).

Line 277-279: In the halophilous scrub, soil CH4 emissions were detected in April, June   and September and the highest CH4 absorption  was observed in February, when the soil was flooded.

Line 279-281: Maximum soil CH4 emissions for the salt meadow were recorded in July   and for the glasswort sward in March or June in flooded and non-flooded soils, respectively,

Line 285: SR and  were

*Discussion*

Line 316-319: The same  was true for  *E. atherica* and *S.patula*.  their maximum photosynthetic rates at La Pletera (29.1 ± 2.4 and 20.8 ± 2.9 μmol CO2 m-2 s-1, respectively) were higher than those previously reported for *E. atherica* (18 μmol CO2 m-2 s-1, Rozema & Diggelen 1991) and for the annual species *Salicornia ramosissima* (14 μmol CO2 m-2 s-1, Pérez-Romero et al. 2018) grown under controlled conditions.

Line 322-324: Studies reporting photosynthetic rates of dominant salt marsh plant species under field conditions are scarce, and the values obtained often diverge substantially from those recorded under controlled conditions.

Line 335-338: Interestingly, photosynthetic rates of the studied species at La Pletera were much lower in autumn than in spring, despite the environmental parameters, such as temperature and soil moisture,  also being favorable  for photosynthesis in autumn, especially in October, where maximum temperature was 21°C and soil VWC was even higher than in March and April.(Pascual, 2022). A possible explanation might be related  to the high accumulation of

ions and soluble carbohydrates  in these species  after a salt stress period,  which occurs in the Mediterranean salt marshes during summer.

Line 346-348: especially in March and May and before sunset, with values of photosynthesis reaching 12 μmol CO2 m-2 s-1 . These values are in agreement with data reported for Californian evergreen species (Saveyn et al., 2010) or for savannah shrubs and trees (Cernusak et al., 2006; Levy and Jarvis, 1998).

Line 357-358: Regarding night respiration rates, the highest values for the four species were recorded in summer (August) and/or autumn (November),  with those found for the green tissues of *S. fruticosa* and *E. atherica* during these months being especially elevated.

Line 362: references are large

Line 364-365: In November, respiration rates were also very high despite that the minimum temperature was much colder (4.6 °C) than in August (22.2 ºC) and similar to February (5.9°C) (Pascual, 2022).

Line 384-386: On the contrary, the sparse vegetation,  of the glasswort sward and the poorly developed root system of its dominant species, *S. patula*, would make  the contribution of roots to soil respiration in this habitat negligible.

Line 437: This would not be the case  in the salt meadow, the most distant habitat from the sea,

---

## Author Comment (AC1)

**Comments R1:**

This is a promising study of carbon emissions and soil mineralization potential from non-tidal salt marshes which offers unique data with which to improve understanding of relationships between gas fluxes, plant physiology, and salinity. Marshes in the region are relatively understudied and the non-tidal system is also poorly known in terms of methane dynamics. The measurement of CO2 exchange from woody plant tissues reveals new insights about the capacity for carbon uptake throughout the plant body in the salt marsh species studied here.

Thank you very much for the feedback on the value of the study.

One major finding is that the salt meadow and halophilous scrub had lower soil mineralization potential but higher soil CO2 emissions than the glasswort sward. This result seems counter-intuitive and I would like to see more discussion to potentially explain how longer term carbon could be sequestered despite the shorter term CO2 fluxes from soils.

We agree that, at first glance, this result may seem counter-intuitive, but it is important to remember that the mineralization quotient is calculated as the ratio between carbon emitted and carbon stored as soil organic carbon (SOC) (Pinzari et al., 1999). In a parallel study, we found significantly higher amounts of carbon in the soils of the halophilous scrub and the salt meadow than in the glassword sward, in accordance with a much higher aboveground, belowground and litter biomass in the former two habitats (Carrasco-Barea et al. 2023). Therefore, despite the higher soil $CO_2$ emissions in the halophilous scrub and the salt meadow, they showed lower mineralization quotients due to their much greater amount of SOC. We have clarified this in the discussion in lines 404-409 (*"Despite the halophilous scrub and the salt meadow had higher soil $CO_2$ emissions than the glasswort sward, they showed lower mineralization quotients due to a much greater amount of SOC (Table S3), which was in accordance with a much higher aboveground, belowground and litter biomass (Carrasco-Barea et al., 2023). Hence, our results would indicate that soils of the halophilous scrub and the salt meadow would have a higher carbon sequestration potential, despite their higher soil carbon emissions".*).

Authors meanwhile found higher CO2 uptake from woody tissues of the plants in the former two habitats. I wonder whether this might contribute to or be related to the higher carbon sequestration rates in those soils?

As commented above, soil carbon sequestration capacity is related to soil $CO_2$ emissions and soil organic carbon content, which is directly linked to the amount of organic matter that arrives to the soil surface, ready to be decomposed and integrated into the soil. The $CO_2$ fluxes from woody stems measure the net $CO_2$ exchange between the woody living surface of plants and the atmosphere, and thus are not directly related to what happens at the soil compartment. Nevertheless, we could speculate that higher $CO_2$ uptake by woody tissues might lead to increased woody tissue biomass, potentially resulting in greater incorporation of more recalcitrant organic matter into the soil. However, this remains speculative, and we have decided not to include it in the discussion.

Another highlight from the study is that relatively large methane emissions were observed despite the salinity of the marshes. Authors partially attribute these high methane fluxes to the influence of low salinity groundwater, which is logical. However, I am concerned that the value of the methane emissions may be over-estimated due to

the long duration of chamber closure (24h). Taking only 2 samples (initial and final) over this 24 period limits the precision of the methane fluxes as well. Authors should take care to interpret their emissions in relative terms (comparison between habitats) rather than drawing comparisons with literature, unless they find studies that have employed similarly long chamber deployments.

*We agree that using different methodologies is a handicap when comparing results from various studies. Thus, we have included a sentence indicating which of the mentioned studies collected samples after 24h of chamber closure as we did, in lines 453-455 ("although it is worth mentioning that only Hirota et al. (2007) took samples after 24h of chamber closure, as it was performed in the present study").*

The data presentation does need to be improved. Please use different symbols or colors to distinguish the marsh habitats or plant species in all data figures. With the current version (all gray), one cannot discern these groups.

*Following your suggestion, we have improved data presentation using different types of lines to distinguish the different habitats and species in all the figures.*

Additional detailed suggestions are attached.

**Review for 2024-1320**

**Abstract**
Line 14: Clarify that H. portulacoides and E. atherica are part of the same habitat (similar to line 85)

*Done.*

**Introduction**
Consider also methylotrophic methanogens which persist in saline environments.

*We have included a sentence about the predominance of methylotrophic methanogens in saline environments in lines 64-67 ("Specifically, aceticlastic and hydrogenotrophic methanogens, with their lower energetic yields, are more susceptible to increasing salinity than methylotrophic methanogens, which explains the predominance of methylotrophic methanogens like Methanohalophilus spp. in hypersaline environments (Mcgenity and Sorokin, 2018)."*

Line 68- How extensive are non-tidal marshes in the Mediterranean? Elsewhere?

*This information has been added to the text in lines 72-74 ("Hence, considering the extensive coverage of non-tidal salt marshes in the Mediterranean Basin, which has been estimated in approximately 19 million hectares (around 2.5% of the total area of the 27 Mediterranean countries and 1 to 2% of wetlands in the world; Geijzendorffer et al., 2018)").*

Line 85-90: Which plant species are C3 vs C4?

*It is specified in lines 91-92 that all species are C3 plants.*

Line 93: Specify the temperature and salinity ranges typical of this region. Are there any salinity differences in the marsh soils between habitats or seasons?

We have included the lowest and the highest mean temperatures in the text (line 98) and we have added a Figure (supplementary material, Figure S2) showing monthly mean daily temperatures and total rainfall for the previous 10 years and for the study year (2017). Besides, we have included the groundwater salinity in section 2.1 (lines 103-105). Soil salinity differences between habitats and seasons (represented as variations in soil electrical conductivity) are shown in section 3.2.1 (Figure 3).

**Section 2.2**
How does severing the plant stems and leaves affect the CO2 fluxes? How long were they stored in the refrigerator?
(Were any measurements done on live, intact plants in the field?)

All measures of net $CO_2$ fluxes from vegetation were conducted in the field using intact and attached plant tissues. We have clarified this in the text in lines 108-109 *("Measurements were performed in the field, using attached living green and woody plant tissues")*. After measures were performed, we collected the measured plant fractions, and we stored them in a fridge until the sampled leaf area was determined (within the next 24h). We have included this information in line 119.

Line 110-111: How was stomatal conductance measured?

We measured stomatal conductance with the same infrared gas analyser (IRGA) used for net $CO_2$ exchange measures (CIRAS-II, PPsystems USA). We have clarified this in the text (line 107).

Line 126-130: Specify make/model of the gas chromatograph- was it using flame ionization detector?

We have specified the model of the gas chromatograph used (Agilent 7890A, Agilent Technologies USA) and that it was connected to a thermal conductivity detector (lines 143-144).

Two methods for soil respiration rates are reported: The soda lime method was used when soils were not flooded. A gas chromatography based method was used when soils were flooded. How was "flooded" defined?
How many measurements were made with each method? (This information will help readers to understand whether flooding frequent or infrequent).

We considered a soil as flooded when it was covered by water. We have included this information in line 140. The number of measures performed with each method are detailed in Table S2. A sentence has been added to inform readers about this Table (lines 154-155).

I am not familiar with the soda lime method. Authors should better support their statement that gas chromatography underestimates CO2 fluxes relative to the soda lime method. With only an initial and final time point, over 24 hours, the fluxes are not very precise. They may be affected by artifacts such as accumulation of pressure or altered temperatures, both of which could influence the gas fluxes measured.

We agree that the statement about gas chromatography has not enough support, and thus we have decided to remove the sentence "it has been observed that gas chromatography can underestimate $CO_2$ emission rates by up to 45% in comparison with the soda-lime method (Lou and Zhou, 2006)". Instead, we have focused on explaining why we chose the

soda lime method, rather than gas chromatography, in order to have an integrative measure of soil $CO_2$ fluxes throughout the whole day (day and night), see lines 151-154 *("Gas chromatography analyses were not used to estimate soil respiration when the soil was not flooded because temperature and humidity variations throughout the day and night could affect the concentration of gas components in the sample (Rochette and Hutchinson, 2005), not being this a problem by using the soda-lime method, which can integrate soil $CO_2$ fluxes over long periods, such as 24h (Keith and Wong, 2006)").* We have also added a sentence clarifying that previous studies have demonstrated that soda lime is a reliable method for estimating soil $CO_2$ fluxes, see lines 128-129 *("This method gives a reliable and integrative measurement of soil $CO_2$ fluxes throughout the whole day (Keith and Wong, 2006)").*

**Results**
Figures: Colors or symbols are needed to distinguish the species represented by each line.

We have improved the data presentation by using different types of lines to distinguish habitats and species in the figures.

Figure 4a: Since these are different methods used to measure CO2 fluxes from flooded vs unflooded soils, the study should not make claims about differences in soil respiration between flooded and exposed conditions. Likewise, authors should omit the flooded data points from Figure 4a to avoid direct comparison with the non-flooded data.

We understand your concern, but we believe this comparison is interesting despite the different methodological approaches applied. For this reason, we have decided to keep these data points in Figure 4a, but, in order to clarify that measurements were performed using two different methods, we have added a sentence in the figure caption and we have highlighted this limitation in the discussion in lines 427-428 *("However, since different methods were used to measure soil respiration in flooded and non-flooded soils, this comparison should be interpreted with caution.").*

**Discussion**
Could the higher photosynthetic rates be related to C4 metabolism in E. atherica (in addition to structural difference in stomata?) Which species are C3 vs C4?

As commented before, all the studied species are C3 plants (lines 91-92).

Line 327: Listing the species in consistent order of water use efficiency would be clearer for the reader

Thank you for your suggestion. We have ordered the species from high to low WUE values (Lines 344-345).

Line 397-399: Avoid direct comparison of flooded and unflooded CO2 fluxes (as mentioned above) due to differences in methods. Authors might rather consider that flooding waters are a known physical barrier to gas exchange.

As commented above, we have highlighted this limitation in the discussion (lines 427-428) and explained the effect that flooding has on the diffusion of $CO_2$ molecules in lines 426-427 *("A reduction of soil $CO_2$ emission to the atmosphere during flooding conditions can be explained by the fact that $CO_2$ molecules diffuse 10000 times slower in water than in air (Kathilankal et al., 2008).")*

Table 1: Which methods were used in the studies on this table? Are they comparable to those in this study?

The most common method used in these studies is gas chromatography for both $CH_4$ and $CO_2$ measures, with the infrared gas analyser being also used to determine $CO_2$ fluxes in one study. To highlight these methodological differences, we have included this information in Table 1. Despite the time period in which the chamber was closed was generally shorter than in our study, the study by Hirota et al. (2007) also kept the chamber closed during 24h. We have added some comments about this to the discussion in lines 420-423 ("*Nevertheless, it should also be noted that the methodology used to determine soil $CO_2$ and $CH_4$ fluxes differs from that generally employed in the studies listed in Table 1, since most of them used gas chromatography for both $CH_4$ and $CO_2$ measurements. Thus, an effect caused by these methodological differences cannot be excluded*"), in lines 427-428 ("*However, since different methods were used to measure soil respiration in flooded and non-flooded soils, this comparison should be interpreted with caution.*") and in lines 453-455 ("*although it is worth mentioning that only Hirota et al. (2007) took samples after 24h of chamber closure, as it was performed in the present study.*").

Discussion of methane lines 401-414: Most of the methane fluxes were positive, and so authors should not mislead the reader by first discussing negative fluxes (indicating consumption). Similarly, the methane emissions did not differ statistically between habitats. Discussion in this section should therefore focus on what might have been similar between habitats and/or how the general magnitude of fluxes falls within the range reported in other marshes.

This part of the discussion has been rewritten following your suggestion (lines 429-455). Thank you for the comment.

Line 425-435: This paragraph about salinity relationships to methane emissions is useful for readers to place this study site and its findings in context. This information about the site salinity should be incorporated into the methods/ site description.

We have included this information within section 2.1 (lines 103-105).

**Conclusions**

Authors should discuss the possible relationship of the high CO2 uptake of woody tissues with the high carbon sequestration potential (as reflected by low mineralization quotients) for the salt meadow and halophylic scrub. Can this help to reconcile the finding of lower soil mineralization quotients despite the high respiration and methane emissions observed?

As we mentioned above, soil carbon sequestration capacity is related to soil $CO_2$ emissions and soil organic carbon content, while $CO_2$ fluxes from woody stems measure the net $CO_2$ exchange between the living woody surfaces of plants and the atmosphere and are therefore not directly related to the soil compartment. Hence, although we could speculate that higher $CO_2$ uptake by woody tissues might lead to increased woody biomass and potentially more organic matter being incorporated into the soil, this remains purely speculative. Therefore, we have decided not to include this speculation in the discussion.

CO2 emissions may be higher in this study than in other previous studies due to the long period of chamber closure (24h) and associated artifacts discussed above.

The soda-lime method has been proved to be a reliable way to integrate daily soil $CO_2$ fluxes (as commented above, see for instance Keith and Wong, 2006). Previous studies have also used 24h of chamber closure, as we have now highlighted in the discussion (lines 453-455) (Hirota et al., 2007). In response to your comments, we have also included several sentences to inform the reader about the different methodologies used in the studies cited throughout the manuscript (lines 420-423 and 427-428).

---

## Author Comment (AC2)

**Comments R2:**

**General comments:**

In a world dealing with climate change, there is a need to better understand all ecosystems. Studies like this one, investigating GHG exchange in understudied ecosystems like non-tidal salt marches are relevant and important. The combination of in-situ CO2 – CH4 soil fluxes with CO2 vegetation fluxes in the different habitats results in interesting insights and a valuable addition to the laboratory studies with controlled conditions previously carried out. The study highlights the seasonal variability and the differences between species well.

Thank you for your positive feedback on the value of the study.

The Materials and Methods sections could be more detailed. Many elements are not mentioned here such as soil information, salinity, more specific climate data, the amount of data points taken and details about calculations are also left out. Possible additions and suggestions are mentioned in the attached file.

We have included information about salinity and climatic data in section 2.1. We have also added more specific climate data (Figure S2), and a table showing mean soil SOC, TN and bulk density parameters obtained for the three studied habitats (Table S3). Detailed information about the samplings performed are shown in the Supplementary Material tables, while further explanation about the mineralization quotient calculations has also been added. Lines where this information has been included are specified in the responses to the specific comments (see below).

The Carbon mineralization quotient is not entirely clearly explained for me in the method section and not much explanation is given in the result and conclusion section. I think more explanation is needed around the mineralization quotient calculation and some discussion is needed around the carbon sequestration potential of the habitats as this result is interesting but not well supported.

The mineralization quotient has been more thoroughly explained in the Material and Methods section and commented in detail in the Discussion section. The lines where this information has been included are specified in the responses to the corresponding specific comments (see below).

The authors mention large discrepancies between measurement methods (GC, soda lime) and between in situ and laboratory experiments, therefore this information should be added when comparing to literature. Especially for the soil fluxes it should be mentioned which method and closure time is used in the literature you compare to. In this study, the closure time of the chamber for the GC method is very long, with only 2 data points (before and at the end of the closure time), so this will seriously influence the fluxes.

We have always compared our results with previous studies of soil carbon fluxes conducted under field conditions. This has been clarified in the caption of Table 1. We have also added information about the methodology used in these previous studies (Table 1), emphasizing the limitations of comparing data obtained by means of different methodologies (lines 420-423 and 427-428). Moreover, we have specified which previous studies used the same chamber closure time (lines 453-455) and we have highlighted the reliability of the soda lime method to estimate integrated soil $CO_2$ fluxes over long time periods, such as the one used in the present study (lines 128-129 and 151-154).

Data presentation can be improved by the inclusion of a table with soil parameters, a table with the main results, a map of the study region and graph of climate data (either in supplementary or in the main text).

We have included all this information as Supplementary Material.

More comments and also some technical corrections and suggestions for readability are included in the attached document.

**Specific comments:**

*Introduction*

Line 68: How extensive are the salt marshes worldwide/in this region and what is the proportion of tital vs non-tidal in this region?

This information has been added to the text in lines 72-74 *("Hence, considering the extensive coverage of non-tidal salt marshes in the Mediterranean Basin, which has been estimated in approximately 19 million hectares (around 2.5% of the total area of the 27 Mediterranean countries and 1 to 2% of wetlands in the world; Geijzendorffer et al., 2018)").*

*Materials and Methods*

Line 79: Authors could add a map of the region

Following your suggestion, we have added a map of the study zone in the Supplementary Material (Fig. S1).

Line 92: Authors could add some climate data of the region both in numbers and a graph. Annually average rainfall, mean temperatures, ... also add some soil data if this is available.

We have included the lowest and the highest mean temperatures in the text (line 98) and we have added a Figure (supplementary material, Figure S2) showing monthly mean daily temperatures and total rainfall for the previous 10 years and for the study year (2017). Besides, we have included the groundwater salinity in section 2.1 (lines 103-105). Soil salinity differences between habitats and seasons (represented as variations in soil electrical conductivity) are shown in section 3.2.1(Figure 3).

How many months is the soil flooded on average? Is this different between the different habitats?

We have included a sentence providing information on the flooding duration of each habitat in lines 101-103 *("The duration of flooding varies among habitats, with the shortest duration in the salt meadow (a few days at most), an intermediate duration in the halophilous scrub (several weeks) and the longest duration in the glasswort sward (ranging from several weeks to several months) (Pascual and Martinoy, 2017)").*

Be sure to add the bulk density, SOC, C and N from the three habitats as these are important in the discussion

We have added a table to the Supplementary Material (Table S3) showing mean SOC, TN and bulk density values for the three studied habitats.

Section 2.2: Is the NER measured in situ with the leaves and tissues attached to the plants?

All measures of net $CO_2$ fluxes from vegetation were conducted in the field using intact and attached plant tissues. We have clarified this in the text in lines 108-109 *("Measurements were performed in the field, using living attached green and woody plant tissues").*

How many leaves and tissues were measured per plant and how many plants were measured per species per session and in total?

The number of plants measured per species and the frequency of samplings, considering the time of the day, are detailed in Table S1. A sentence has been added to inform readers about this Table (lines 116-117).

Section 2.3: I'm don't know the soda lime method very well, but was the amount of soda lime needed for the chambers tested to be sure of complete absorption of the efflux or was this based this on previous literature?

The amount of soda lime required for this study was previously tested by one of the coauthors in earlier works (unpublished data).

I also think the statement starting in line 137 should be more nuanced or better supported, as in the same reference (Lou and Zhou, 2006), there is also indicated "The method (soda lime) tends to overestimate soil CO2 efflux in its low range and underestimate it in its high range compared-with dynamic methods (Yim *et al*. 2002). The technique can potentially underestimate soil surface CO2 effluxes by 10 to 100% (Norman *et al*. 1992, rochette *et al*. 1992, Haynes and Gower 1995, Nay *et al*. 1994)."

The sentence used in this article is based on the sentence from (Lou and Zhou, 2006) "The GC method can potentially underestimate the rate of soil CO2 fluxes in comparison with other methods by up to 45% (Knoepp and Vose 2002)." But if you continue to Knoepp and Vose 2002, it seems to me that the SODA method used in this study is not that good either compared to the other methods and underestimates the CO2 emissions even more than the GC method.

We agree that the statement about gas chromatography has not enough support, and thus we have decided to remove the sentence "it has been observed that gas chromatography can underestimate $CO_2$ emission rates by up to 45% in comparison with the soda-lime method (Lou and Zhou, 2006)". Instead, we have explained why we chose to use the soda-lime method, rather than gas chromatography, to measure soil $CO_2$ fluxes when the soil was not flooded, see lines 151-154 *("Gas chromatography analyses were not used to estimate soil respiration when the soil was not flooded because temperature and humidity variations throughout the day and night could affect the concentration of gas components in the sample, not being this a problem by using the soda-lime method, which can integrate soil $CO_2$ fluxes over long periods, such as 24h (Keith and Wong, 2006)").*

In this study serious underestimation is however a possible issue as the closure time is 24h. This is a long time in which saturation in the headspace can occur. The accumulation of the gas inside the chamber can limit the further emission. However this underestimation is not linked to the GC method but rather to the closure time of the chamber.

Previous studies (see for instance Keith and Wong, 2006) support the reliability of the soda-lime method for measuring soil $CO_2$ emission after long periods of chamber closure (such as 24h). One study cited in the text also kept the chamber closed for 24h (Hirota et al., 2007), although measurements were not performed using the soda lime methodology.

Mention how many measurements were taken in flooded state and how many in non-flooded state and how this is different for different habitats.

The number of measurements performed in flooded and non-flooded soils for each habitat is detailed in Table S2. We have added a sentence to let readers know (lines 154-155).

*Wf* and *Wi* were estimated from volumetric concentration (%) considering the air volume inside the chamber in each sampling date. Is it meant here inclusion of temperature and pressure measurement from the chamber on the sampling date to transform the ppm/ppb measurements from the GC to g CO2/CH4? If yes, which temperature and pressure is used?

No, we did not include temperature and pressure in the calculations. We converted the volumetric concentration (ml $CO_2$/100 ml air) obtained from the GC results to mg $CO_2$ using the chamber volume and $CO_2$ density. We estimated the amount of $CO_2$ (in millilitres) within the chamber and then we converted these ml to g $CO_2$ by multiplying by the $CO_2$ density.

Line146: very small comment but the unit of SMF is g CH4 m-2 d-1 here but later on the unit mg CH4 m-2 d-1 is consistently used.

Thank you. We changed the units in the $CH_4$ flux equation to mg $CH_4$ $m^{-2}$ $d^{-1}$.

Line 164: The bulk density of soil is never mentioned before, so authors could add the values. Also mention the SOC values used (see also previous comment on line 92).

As previously mentioned, we have added Table S3 in Supplementary Material, which provides information on SOC, TN and bulk density for the same plots where we performed the carbon flux measurements.

I'm not familiar with the carbon mineralization quotient. How were the C_CO2 and C_CH4 values calculated?

The carbon content of $CO_2$ (C_$CO_2$) and $CH_4$ (C_$CH_4$) was calculated using the atomic and molecular weights of these molecules. For example, to calculate C_$CO_2$: from X grams of $CO_2$, we estimated the amount of carbon in these X grams by considering that 44 g of $CO_2$ have 12 g of C. Thus, we multiplied the grams of $CO_2$ emitted by 12 and we divided by 44. C_CH4 values were calculated similarly.

How was the transformation from the unit of "g CH4 m-2 d-1" and "g CO2 m-2 d-1" to "mg C g soil-1 d-1"?

The calculation of mg C is explained in the previous response. To convert emissions per unit area to emissions per grams of soil, the volume of soil under the chamber was first estimated by multiplying the area of the chamber by 20 cm, which was the soil depth considered. Then, using the estimated soil volume and bulk density (g soil/$cm^{-3}$), we calculated the grams of soil under the chamber (i.e. the soil from which the carbon emission measured comes from).

We added information about these calculations in lines 179-185 (*"C_$CO_2$ and C_$CH_4$ were calculated multiplying the amount of $CO_2$ and $CH_4$ emitted by 12/44 and 12/16, respectively, being 12 the molecular weight of carbon, 44 the molecular weight of $CO_2$, and 16 the molecular weight of $CH_4$. SOC values were taken from previous measurements performed*

*in July 2015 and 2016 in the same experiment (Table S3), after observing that these values exhibited stability and remained constant over the studied years (Carrasco-Barea et al., 2023). To convert emissions per unit area to emissions per grams of soil, we estimated the volume of soil beneath the chamber by multiplying the chamber area by the considered soil depth (20 cm), and then multiplying this volume by soil bulk density (g soil cm$^{-3}$).")*

**Results**

Concerning the correlation between SR and SOC and TN found in July: there is one value of SOC and TN for each habitat, so three in total? Or is there a value for each plot?

To avoid pseudoreplication, we used the mean of SOC and TOC for each habitat (n=5 per habitat), with each habitat having a single representative value.

**Discussion**

The authors could add a table (maybe in supplementary) with mean/min/max values of water use efficiency, photosynthetic rates, ...

Following your suggestion, we have added a table to the Supplementary Material with mean/min/max values of instantaneous net $CO_2$ exchange rates (NER) of vegetation (Table S4), another with stomatal conductances and intrinsic water use efficiencies (Table S5) and a third one with carbon fluxes and mineralization quotients of non-flooded soils (Table S6).

*Section 5.2.2.*

The soil C and N content is put forward as possible explanation in line 372 and 377 for the higher SR in HS and SM than in GS, so it think the values of SOC and TN should be added in the paper (maybe in supplementary).

As commented before, we have added a table with the mean values of SOC, TN and bulk density to the Supplementary Material (Table S3).

Line 375-378: Accordingly, a positive correlation was found between July SR and SOC or TN content at the halophilous scrub and the salt meadow of La Pletera salt marsh, since these two habitats had higher content of SOC and TN than the glasswort sward (Carrasco-Barea et al., 2023).

Was there only a positive correlation found between SR and SOC and TN for HS and SW because in the results this is not specified (see my one remark in the section Results above). Also if you have one value of SOC or TN per habitat, then how is a correlation per habitat found, which brings me back to my previous question in Result section?

Maybe this needs to be rephrased, framing that the positive correlation found between July SR and SOC or TN content across the habitats together with the fact that HS and SM have higher SOC an TN underpin the statement that the differences in SR might be related to the soil C and/or N content.

Thank you for your comment, since there was an error in the redaction of these sentence. As you noticed, it seems that we did a correlation per each habitat, when, in fact, we did a correlation considering one value (the mean) per habitat. We rephrased the sentence in order to clarify this as follows (lines 393-396): *"In our study, a positive correlation was found between July SR and SOC or TN content reinforcing the idea that differences in SR among*

*habitats would be related with the higher SOC and TN content found in the halophilous scrub and the salt meadow compared to the glasswort sward (Table S3)".*

Line 394: can the "occasional tide" be more specific. Are we talking about flooding during several weeks in specific months or also small occasional floods once every week?

We have clarified the meaning of "occasional tide", including the frequency with which this salt marsh is typically flooded and the shortest and longest flood durations across the different habitats in lines 417-418 *("In tidal salt marshes, flooding occurs once or twice every day, while it is occasional at La Pletera (1-2 times per year, remaining the soil flooded from some days in the salt meadow to several weeks or even months in the glasswort sward) (Pascual and Martinoy, 2017).").* This information has also been added to the Materials and Methods section (lines 101-103).

Line 406-413: The way this section is written makes it seem like there is mainly absorption of methane and then some sudden high emission peaks, while it is actually mainly emissions that are measured. Maybe a percentage of negative fluxes to total fluxes can be given.

This part of the discussion has been rewritten. We have now focused the discussion on the seasonal changes and the comparison between habitats and with previous studies performed in other marshes (lines 429-455).

Line 411-413: "In the glasswort sward, peaks in CH4 emissions were observed both when the soil was not flooded (110± 59 mg CH4 m-2 d-1) and when it was flooded (131 ± 45 mg CH4 m-2 d-1), highlighting that methane oxidation in the overlying water column would not be happening."

I'm not sure that emissions during both non flooded and flooded states prove that there is no oxidation in the overlying water column. It states that the methane oxidation is rather limited or actually that the methane oxidation is not a big factor as the net emission is still large.

This part of the discussion was removed following a suggestion from reviewer 1.

Line 439-442: Not much is said about this mineralization quotient. Is there an explanation for the higher sequestration potential of the HS and SM or the lower sequestration potential of GS? I assume that the SOC amount of GS is very low compared to the SOC amount of HS and SM (be sure to put these values in).

As commented above, we have added SOC values to Table S3 and discussed the results in more detail in the discussion, relating the mineralization quotient to the SOC values in lines 404-409 *("Despite the halophilous scrub and the salt meadow had higher soil $CO_2$ emissions than the glasswort sward, they showed lower mineralization quotients due to a much greater amount of SOC (Table S3), which was in accordance with a much higher aboveground, belowground and litter biomass (Carrasco-Barea et al., 2023). Hence, our results would indicate that soils of the halophilous scrub and the salt meadow would have a higher carbon sequestration potential, despite their higher soil carbon emissions".).*

Table1. The authors mention large discrepancies between measurement methods and between in situ or laboratory experiments, therefore this information can maybe be added in the table.

We have added to Table 1 the methodology used in each study and we have specified in the table caption that all the studies cited were conducted *in situ*.

I would suggest to move this table to supplementary material and instead incorporate some tables with the data gathered from this study. For readers who want to quickly scan the paper, a table with the mean results of the study would be handy.

We think that Table 1 helps readers to quickly compare present data with data from previous studies and easily locate values discussed in the text. For this reason, we have decided to keep it in the main manuscript. However, following your suggestion, we have included three tables (Tables S4, S5 and S6) with the main results, although they have been added to the Supplementary Material to avoid redundancy with the Figures in the manuscript.

**Technical corrections:**

***Abstract***

Line 19-21: Regarding the studied habitats, the halophilous scrub and the salt meadow showed higher soil $CO_2$ emissions than the glasswort sward,  and the overall emissions were higher than those previously reported for tidal salt marshes.

Done.

***Introduction***

Line 29:  compared to the atmospheric concentration

Done (line 31).

Line 30-32: In this context of continuous global warming, ecosystems play an important role in global climate regulation,  Therefore, it is essential to determine net emissions of greenhouse gases of ecosystems to estimate their effects on global warming.

Done (lines 31-33).

Line 34: (Laffoley & Grimsditch, 2009) is not present in the references.

We have included this citation in the reference's list.

Line 35-39: Previous studies on the photosynthetic capacity of salt marsh halophytic species have mainly focused on the effect of salinity on photosynthetic rates,  and most of these studies  were performed under controlled conditions (Davy et al., 2006; Duarte et al., 2014; Kuramoto and Brest, 1979; Nieva et al., 1999; Pearcy and Ustin, 1984; Redondo-Gómez et al., 2007) and less frequently under field conditions (Drake, 1989; Maricle and Maricle, 2018; Warren and Brockelman, 1989).

Done (lines 36-40).

Line 45-47: Photosynthetic rates also depend on abiotic factors, such as light, temperature, flooding regime, salinity or nutrient availability (Drake, 1989; Huckle et al., 2000). In

general it is assumed that the highest plant photosynthetic activity occurs during the hours of the day with the highest solar radiation (midday).

Done (lines 45-48).

Line 55-56: In salt marshes, flooding also has a major effect on CO2 and CH4 emissions, since it determines which process, aerobic respiration or anaerobic metabolism, prevails.

Done (lines 56-57).

Line 60-62: Nevertheless, , generally soil CH4 emissions are negatively affected by salinity (Bartlett and Harriss, 1993; Livesley and Andrusiak, 2012; Poffenbarger et al., 2011), since in saline environments sulphate-reducing bacteria use to compete with methanogens for energy sources, and consequently disfavor and even inhibit methane production.

Done (lines 61-64).

Line 65-68: Despite the importance  of soil carbon fluxes  in climate regulation, few studies have characterized these fluxes in Mediterranean salt marshes (Wang, 2018), and, to our knowledge, not one study has been performed in non-tidal salt marshes (tides range from 0.1 to 1 m, in contrast to 1-10 m of tidal salt marshes) of the Mediterranean Basin.

Done (69-72).

**Materials and Methods**

Line 79-80: The study was performed at La Pletera, a coastal Mediterranean non-tidal salt marsh located in the north of the river mouth of the Ter  in the municipality of Torroella de Montgrí.

Changed to "... in the north of the mouth of the Ter river ..." (lines 84-85).

Line 86-87: being all these species C3 -> , all these species being C3

Done (lines 91-92).

Line 136: the air volume inside the chamber  on each sampling date.

Done (lines 149-150).

Line 151: air volume inside the chamber  on each sampling date.

Done (line 167).

Line 157 : stored in the soil (SOC)  at/above a certain depth

Done (lines 173-174).

**Results**

Line 203: Differences among species in instantaneous  from green tissues

Done (line 222).

Line 205: after sunrise with no significant differences with *S. fruticosa* in April Fig. 1a),

Done (lines 223-224).

Line 232: During most of the year, *E. atherica* showed the highest values of  at midday

Done (line 151).

Line 272:  for non-flooded soils of the three

Done (line 291).

Line 273-274:  $CO_2$ emissions were remarkably lower when soils were flooded.

Done (line 292).

Line 276-277: Remarkably high peaks of soil $CH_4$ emissions were recorded in the three habitats,  but also negative values,  were  observed (Fig. 4b).

Done (lines 295-296).

Line 277-279: In the halophilous scrub, soil $CH_4$ emissions were detected in April, June  and September,  and the highest $CH_4$ absorption  was observed in February, when the soil was flooded.

Done (lines 296-297).

Line 279-281: Maximum soil $CH_4$ emissions for the salt meadow were recorded in July for the salt meadow, and for the glasswort sward in March or June in flooded and non-flooded soils, respectively.

Done (lines 298-299).

Line 285: SR and  were

Done (line 303).

**Discussion**

Line 316-319: The same  was true for *E. atherica* and *S.patula*.  their maximum photosynthetic rates at La Pletera (29.1 ± 2.4 and 20.8 ± 2.9 µmol $CO_2$ m-2 s-1, respectively) were higher than those previously reported for *E. atherica* (18 µmol $CO_2$ m-2 s-1, Rozema & Diggelen 1991) and for the annual species *Salicornia ramosissima* (14 µmol $CO_2$ m-2 s-1, Pérez-Romero et al. 2018) grown under controlled conditions.

Done (lines 335-338).

Line 322-324: Studies reporting photosynthetic rates of dominant salt marsh plant species under field conditions are scarce, and the values obtained often diverge substantially from those recorded under controlled conditions.

Done (lines 341-343).

Line 335-338: Interestingly, photosynthetic rates of the studied species at La Pletera were much lower in autumn than in spring, despite the environmental parameters, such as temperature and soil moisture,  also being favorable  for photosynthesis in autumn,

(especially in October, where maximum temperature was 21°C and soil VWC was even higher than in March and April(Pascual, 2022). A possible explanation might be related  to the high accumulation of ions and soluble carbohydrates  in these species  after a salt stress period,  which occurs in the Mediterranean salt marshes during summer.

Done (lines 354-359).

Line 346-348: especially in March and May and before sunset, with values of photosynthesis reaching 12 μmol CO2 m-2 s-1. These values are in agreement with data reported for Californian evergreen species (Saveyn et al., 2010) or for savannah shrubs and trees (Cernusak et al., 2006; Levy and Jarvis, 1998).

Done (lines 365-367).

Line 357-358: Regarding night respiration rates, the highest values for the four species were recorded in summer (August) and/or autumn (November),  with those found for the green tissues of *S. fruticosa* and *E. atherica* during these months being especially elevated.

Done (lines 375-377).

Line 362: references are large

We have reduced the font size.

Line 364-365: In November, respiration rates were also very high despite that the minimum temperature was much colder (4.6 °C) than in August (22.2 °C) and similar to February (5.9°C) (Pascual, 2022).

Done (lines 382-383).

Line 384-386: On the contrary, the sparse vegetation, (which is only alive during few months) of the glasswort sward and the poorly developed root system of its dominant species, *S. patula*, would make  the contribution of roots to soil respiration in this habitat negligible.

Done (lines 402-404).

Line 437: This would not be the case  in the salt meadow, the most distant habitat from the sea,

Done (lines 450-451).

---

## Referee Report (RR1)

**Major revision iteration of article egusphere-2024-1320:**

**Seasonal carbon fluxes from vegetation and soil in a Mediterranean non-tidal salt marsh**

This study investigated GHG exchange in understudied ecosystem, namely non-tidal salt marches. The study combines in-situ $CO_2$ – $CH_4$ soil fluxes in both flooded and non-flooded periods in different habitats, with CO2 vegetation fluxes from dominant species. The study highlights potential drivers behind the differences between species and de difference between seasons

The authors clearly took the advice of the previous referees well in consideration. I appreciate the inclusion of the map with the study site location, the table with soil data as well as the graphs with climate data in the supplementary material. The data representation in the article improved and the inclusion of extra tables with mean values in supplementary promotes easy data access.

I also appreciate the additions and corrections carried out in the article as wel as the better framed results, taking into account the possibility of discrepancies due to the comparison of different methods used in this study. The authors also nuance their results by adding that the discrepancies with other studies may be due to the fact that they use different methods compared to other studies. The overall readability of the article improved a lot. The method section more clearly explains the used variables and equations and also the discussion section is now clearer, nicely highlighting the important results.

Some more rather small corrections could be made to improve readability:

Line 33: You don't mention NO2 and SF6 in the text so maybe the authors don't need to reference it here. Also is NO2 meant or is it N2O?

Line 34: compared to the atmospheric ...

Line 62: Since it determines which process ...

Line 76: to our knowledge, not one study has been performed in ...

Line 110: The salinity of the water table is around 0.86 ‰, being typical the sea water intrusion during summer, which moves the saltwater wedge inland, increasing groundwater salinity until levels similar to those of the sea (approximately 32 ‰) (Menció et al., 2017)

[Figure]

    The salinity of the water table is around 0.86 ‰. During summer,  sea water intrusion typically happens, which moves the saltwater wedge inland and increases groundwater salinity until levels similar to those of the sea (approximately 32 ‰) (Menció et al., 2017)

Line 125: After measuring CO2 fluxes in the field, the used plant fractions were collected and stored in a fridge until sampled area was determined in the laboratory (within the next 24h).

Line 162: .... temperature and humidity variations throughout the day and night could affect the concentration of gas components in the sample (Rochette and Hutchinson, 2005), not being this a problem by using the soda-lime method, which can integrate soil CO2 fluxes over 165 long periods,

such as 24h (Keith and Wong, 2006). The number of flooded and non-flooded plots, as well as the method use in every sampling day are detailed in Table S2.

||
v

... temperature and humidity variations throughout the day and night could affect the concentration of gas components in the sample (Rochette and Hutchinson, 2005). This is not a problem when using the soda-lime method, which therefore can integrate soil CO2 fluxes over long periods, such as 24h (Keith and Wong, 2006).

Line 192: "C_CO2 and C_CH4 were calculated by multiplying the amount of CO2 and CH4 emitted with 12/44 and 12/16, respectively, with 12 the molecular weight of carbon, 44 the molecular weight of CO2, and 16 the molecular weight of CH4. To convert emissions per unit area to emissions per grams of soil, we estimated the volume of soil beneath the chamber by multiplying the chamber area by the considered soil depth (20 cm), and then multiplying this volume by soil bulk density (g soil cm-3 ). SOC values were taken from previous measurements performed in July 2015 and 2016 in the same experiment 195 (Table S3), after observing that these values exhibited stability and remained constant over the studied years (Carrasco-Barea et al., 2023).

Line 254 ... and neither between night NER and minimum air temperature

Line 284: The highest  were registered during ...

➔ Introduce abbreviations only once, the first time they are mentioned.

Line 286: Significant differences in the seasonal  (VWC) of the soil were ...

➔ Introduce abbreviations only once, the first time they are mentioned.

Line 289:  was significantly higher in the ...

➔ Introduce abbreviations only once, the first time they are mentioned.

Line 400: ... were also very high despite that the minimum temperature was ....

Line 436: ..., while it is occasional at La Pletera; only 1-2 times per year, leaving the soil flooded for some days in the salt meadow to several weeks or even months in the glasswort sward.

Line 439: Nevertheless, it should  be noted that the methodology used to determine soil CO2 and CH4 fluxes in this study, differs from  the one generally employed in the studies listed in Table 1, since most of them used gas chromatography for both CH4 and CO2 measurements. Thus, an effect caused by these methodological differences cannot be excluded.

Line 443: Previous studies under field (Kathilankal et al., 2008; Moffett et al., 2010) or laboratory (Jones et al., 2018; Wang et al., 2019) conditions support a negative effect of flooding on

soil CO2 emissions, as it has been found at La Pletera. At La Pletera, the reduction of soil CO2 emissions  during flooding conditions can be explained by the fact that CO2 molecules diffuse 10000 times slower in water than in air (Kathilankal et al., 2008). However, since different methods were used to measure soil respiration in flooded and non-flooded soils, this comparison should be interpreted with caution

Line 469: Despite no soil anaerobic conditions,  necessary for the growth of methanogens , would be expected during summer because of the low soil VWC at La Pletera salt ...

Line 492: although it is worth mentioning that only Hirota et al. (2007) took samples after 24h of chamber closure, as  was  done in the present study. .

6 Conclusions -> 5 Conclusions

---

## Author Response (AR2)

**Answers to the reviewer comments of the manuscript egusphere-2024-1320 after the major revisions iteration**

Here we show the answers to the reviewer comments of our manuscript (egusphere-2024-1320) after the major revision iteration. The comments of the reviewer are marked in black, the answer to the comments in blue and the changes that we have made in green. Line numbers refer to the revised manuscript where the changes are marked.

**Comments Reviewer:**

This study investigated GHG exchange in understudied ecosystem, namely non-tidal salt marches. The study combines in-situ $CO_2$ – $CH_4$ soil fluxes in both flooded and non-flooded periods in different habitats, with CO2 vegetation fluxes from dominant species. The study highlights potential drivers behind the differences between species and de difference between seasons

The authors clearly took the advice of the previous referees well in consideration. I appreciate the inclusion of the map with the study site location, the table with soil data as well as the graphs with climate data in the supplementary material. The data representation in the article improved and the inclusion of extra tables with mean values in supplementary promotes easy data access.

I also appreciate the additions and corrections carried out in the article as wel as the better framed results, taking into account the possibility of discrepancies due to the comparison of different methods used in this study. The authors also nuance their results by adding that the discrepancies with other studies may be due to the fact that they use different methods compared to other studies. The overall readability of the article improved a lot. The method section more clearly explains the used variables and equations and also the discussion section is now clearer, nicely highlighting the important results.

Some more rather small corrections could be made to improve readability:

Line 33: You don't mention NO2 and SF6 in the text so maybe the authors don't need to reference it here. Also is NO2 meant or is it N2O?

We have changed these references by citing them as reports instead of web page, as it is recommended by the authors

(see the citation recommendations in:)

https://gml.noaa.gov/ccgg/trends_doi.html

https://gml.noaa.gov/ccgg/trends/global.html

In this way, the name of the authors appears in the citation instead of the tittle.

That is, we have eliminated (Trends in globally-averaged CO2 determined from NOAA Global Monitoring Laboratory measurements.; Trends in globally-averaged CH4, NO2, and SF6 determined from NOAA Global Monitoring Laboratory measurements.) and we have added "(Lan et al., 2023b, a)" (line 30).

We have also corrected the N2O since NO2 was wrong (which appears in title of the report in the reference list).

Line 34: compared to the atmospheric …

Done (line 31).

Line 62: Since it determines which process …

Done (line 58).

Line 76: to our knowledge, not one study has been performed in …

Done (line 71).

Line 110: The salinity of the water table is around 0.86 ‰, being typical the sea water intrusion during summer, which moves the saltwater wedge inland, increasing groundwater salinity until levels similar to those of the sea (approximately 32 ‰) (Mencló et al., 2017)

||

v

The salinity of the water table is around 0.86 ‰. During summer, sea water intrusion typically happens, which moves the saltwater wedge inland and increases groundwater salinity until levels similar to those of the sea (approximately 32 ‰) (Mencló et al., 2017)

Done (lines 104-106).

Line 125: After measuring CO2 fluxes in the field, the used plant fractions were collected and stored in a fridge until sampled area was determined in the laboratory (within the next 24h).

Done (line 119).

Line 162: …. temperature and humidity variations throughout the day and night could affect the concentration of gas components in the sample (Rochette and Hutchinson, 2005), not being this a problem by using the soda-lime method, which can integrate soil CO2 fluxes over 165 long periods, such as 24h (Keith and Wong, 2006). The number of flooded and non-flooded plots, as well as the method use in every sampling day are detailed in Table S2.

||

v

… temperature and humidity variations throughout the day and night could affect the concentration of gas components in the sample (Rochette and Hutchinson, 2005). This is not a problem when using the soda-lime method, which therefore can integrate soil CO2 fluxes over long periods, such as 24h (Keith and Wong, 2006).

Done (lines 154-155).

Line 192: "C_CO2 and C_CH4 were calculated by multiplying the amount of CO2 and CH4 emitted with 12/44 and 12/16, respectively, with 12 the molecular weight of carbon, 44 the molecular weight of CO2, and 16 the molecular weight of CH4. To convert emissions per unit area to emissions per grams of soil, we estimated the volume of soil beneath the chamber by multiplying the chamber area by the considered

soil depth (20 cm), and then multiplying this volume by soil bulk density (g soil cm-3 ). SOC values were taken from previous measurements performed in July 2015 and 2016 in the same experiment 195 (Table S3), after observing that these values exhibited stability and remained constant over the studied years (Carrasco-Barea et al., 2023).

Done (lines 182-186).

Line 254 … and neither between night NER and minimum air temperature

Done (line 243).

Line 284: The highest  were registered during …

→ Introduce abbreviations only once, the first time they are mentioned.

Done (line 274).

Line 286: Significant differences in the seasonal  of the soil were …

→ Introduce abbreviations only once, the first time they are mentioned.

Done (lines 276-277).

Line 289:  was significantly higher in the …

→ Introduce abbreviations only once, the first time they are mentioned.

Done (line 279).

Line 400: … were also very high despite that the minimum temperature was ….

Done (line 385).

Line 436: …, while it is occasional at La Pletera; only 1-2 times per year, leaving the soil flooded for some days in the salt meadow to several weeks or even months in the glasswort sward.

We have made these changes, but we also changed "in the salt meadow  several weeks…" for "in the salt meadow and several weeks…" (lines 420-421).

Line 439: Nevertheless, it should  be noted that the methodology used to determine soil CO2 and CH4 fluxes in this study, differs from  the one generally employed in the studies listed in Table 1, since most of them used gas chromatography for both CH4 and CO2 measurements. Thus, an effect caused by these methodological differences cannot be excluded.

Done (lines 423-426).

Line 443:  Previous studies under field (Kathilankal et al., 2008; Moffett et al., 2010) or laboratory (Jones et al., 2018; Wang et al., 2019) conditions support a negative effect of flooding on soil CO2 emissions, as it has been found at La Pletera. At La Pletera, the reduction of soil CO2 emissions  during flooding conditions can be explained by the fact that CO2 molecules diffuse 10000 times slower in water than in air (Kathilankal et al., 2008). However, since different methods were

used to measure soil respiration in flooded and non-flooded soils, this comparison should be interpreted with caution

Done (lines 427-431).

Line 469: Despite no soil anaerobic conditions,  necessary for the growth of methanogens, would be expected during summer because of the low soil VWC at La Pletera salt …

Done (line 436).

Line 492: although it is worth mentioning that only Hirota et al. (2007) took samples after 24h of chamber closure, as  was  done in the present study. .

Done (line 457).

6 Conclusions -> 5 Conclusions

Done (line 468).